# Functional Foods: An Approach to Modulate Molecular Mechanisms of Alzheimer’s Disease

**DOI:** 10.3390/cells9112347

**Published:** 2020-10-23

**Authors:** Anna Atlante, Giuseppina Amadoro, Antonella Bobba, Valentina Latina

**Affiliations:** 1Institute of Biomembranes, Bioenergetics and Molecular Biotechnologies (IBIOM)-CNR, Via G. Amendola 122/O, 70126 Bari, Italy; a.bobba@ibiom.cnr.it; 2Institute of Translational Pharmacology (IFT)-CNR, Via Fosso del Cavaliere 100, 00133 Rome, Italy; giusy.amadoro@gmail.com; 3European Brain Research Institute (EBRI), Viale Regina Elena 295, 00161 Rome, Italy; valentina.latina80@gmail.com

**Keywords:** Alzheimer, nutraceuticals, brain, cell death, health, diet, gut microbiota, epigenetics

## Abstract

A new epoch is emerging with intense research on nutraceuticals, i.e., “food or food product that provides medical or health benefits including the prevention and treatment of diseases”, such as Alzheimer’s disease. Nutraceuticals act at different biochemical and metabolic levels and much evidence shows their neuroprotective effects; in particular, they are able to provide protection against mitochondrial damage, oxidative stress, toxicity of β-amyloid and Tau and cell death. They have been shown to influence the composition of the intestinal microbiota significantly contributing to the discovery that differential microorganisms composition is associated with the formation and aggregation of cerebral toxic proteins. Further, the routes of interaction between epigenetic mechanisms and the microbiota–gut–brain axis have been elucidated, thus establishing a modulatory role of diet-induced epigenetic changes of gut microbiota in shaping the brain. This review examines recent scientific literature addressing the beneficial effects of some natural products for which mechanistic evidence to prevent or slowdown AD are available. Even if the road is still long, the results are already exceptional.

## 1. Introduction

Fruits and vegetables have risen to the honors of healthy foods since they have been considered valuable antagonists of the aging process, also able to help to fight age-associated diseases. These findings have been confirmed by many epidemiological studies and every specialist, oncologist, cardiologist, angiologist, neurologist or diabetologist on their own, shares this position when recommend consuming at least five portions a day of both fruit and vegetables [1].

It is more than a diet to follow, it is a lifestyle playing an important role in the management of many diseases as well as in the likelihood of having cancer, heart disease, diabetes and others of our most feared diseases. The same goes for the prevention of Alzheimer’s disease (AD).

AD is an age-related disease and represents a common risk in elderly. The diagnosis of AD before age 65 is rare and seems to occur only to a small percentage of people (2–5% of all cases) harboring genetic mutations in relevant genes [2]. However, AD tends to run in families, suggesting a genetic link on one side, but urging to go beyond genetics on other side. Researchers speculate that genetic risk factors can be responsible only for one third of how our brain changes with age [3,4]. The other two thirds of non-genetic factors may depend on lifestyles and the surrounding environment. That is, in other words: we cannot ignore our genetics, our parents or the passing time, but we can control some other factors.

Our diet potentially contributes to the onset of a disease. Eating healthy helps to prevent AD, but a diet that will most likely keep Alzheimer’s away is almost the same that will keep your heart healthy, low cholesterol level, cancer at bay and your glucose levels balanced. In a broad sense: how to eat to live up to 100 years [5,6].

What has been said so far could explain why Popeye is still going strong at almost 90 years of age: eating spinach, cabbage and other green leafy vegetables seems to slow down cognitive decline. In fact, green leafy vegetables as well as fruits appear to have an outsized impact, but other foods also show promising results in protecting our brain. Of course, this is not the place to list detailed dietary recommendations to improve anyone’s health, but it is beyond any doubt that specific dietary recommendations for AD patients are becoming more and more widespread [7,8,9].

The brain is highly susceptible to oxidative damage due to its high metabolic activity and to the presence of abundant oxidizable material, such as the lipid milieu of the myelin membrane typical of neuronal cells. Therefore, antioxidant foods could have positive effects on neural function [10,11,12]. About that, various types of berries have been shown to have strong antioxidant properties due to the presence of tannins, anthocyanins and phenols which are capable of increasing the plasticity of the hippocampus, thus promoting learning and memory performance. Alpha lipoic acid, which is found in vegetables such as spinach and broccoli, is also important for maintaining the energy homeostasis of mitochondria and improving cognitive functions. More, green and black tea, both rich in antioxidants, contain epigallocatechin gallate, which has been shown to indirectly reduce the formation of the amyloid plaques, typical hallmarks of AD. Even eggs, as primary source of numerous nutrients including vitamin B6, vitamin B12, choline and folic acid, may help us to maintain the brain healthy [13,14,15,16,17,18]. Turmeric, a CUR-rich spice responsible for the yellow color of curry, reduces memory deficits caused by AD by blocking the formation of amyloid plaques. Red wine would be more effective than white wine because it contains more resveratrol (RSV) [19,20,21]. Coconut oil could help to prevent AD, in addition to EVOO, the typical component of the Mediterranean diet (MD) [22,23]. The undisputed role of MD has long been known but, recently, it has become even more famous being able to positively modulate the gut microbiome. A new study published last February in the journal Nutrition [24] found that MD, administered to elderly subjects, alters the microbiome and increases the abundance of specific taxa that seem to correlate with improved cognitive function and decreased inflammatory markers.

It was Stephen Defelice, chairman of the American organization known as Foundation for Innovation in Medicine, who first coined the term ‘nutraceutical’, a word derived by the crasis between “nutrition” and “pharmaceutical”, to indicate those components of food which not only have beneficial effects on our health but can also be employed in therapeutic paths to lower the risk of many diseases in humans [25,26].

Nutraceuticals, once absorbed and assimilated in the gut, increase our resistance to harmful environmental agents and therefore behave like drugs with preventive and/or adjuvant action in the healing phase [27]. Mainly present in fruits and vegetables, they are abundantly produced by plants as a defense mechanism against insects and as such are of high value to humans due to the beneficial effects that can be exploited even at low dosage [28,29].

An interesting aspect of the matter is that, although the molecular mechanisms underlying the effect of diet on epigenetics are still poorly understood, an increasing number of innovative studies points to the exciting possibility that the effects of diet on mental health may be passed on through generations. The possibility for a dietary component to modulate non-genetic events, that still cause a potentially heritable phenotypic change, opens a sort of therapeutical route which may be helpful in the management, if not just in the control, of the disease progression [30,31]. Thus, the individual’s risk of suffering from diabetes is increased if paternal grandparents have grown up in times of food abundance rather than shortage.

Following a correct diet is essential to keep fit, but also to preserve our mental health, i.e., improve memory, cognitive functions, intelligence and sharpness.

## 2. Nutraceuticals: Overview

That foods are able to prevent disease is a belief proclaimed by our ancestors over the centuries. Since ancient times, the wise Solomon said “Eat honey, my son, because it is good” (Old Testament, proverb 24:13), stressing the primary importance of honey in nutrition, as a remedy for health problems. Similarly ginseng, whose use in China dates back to 2000 years ago, has always been considered a traditional drug as well as cinnamon that in Ancient Egypt was considered more valuable than gold [27].

Nowadays, the quality of life in terms of income, expenditure and lifestyle has improved a lot on the thrust of economic development. However, eating habits represent the Achilles’ heel of any lifestyle change. Scientific evidence increasingly supports the idea that diet [32]: (i) has a great influence on health in the later stages of life, (ii) is an important modifiable factor for disease prevention and (iii) may minimize complications of the disease.

The worldwide acceptance of these assumptions has seen a rise in the market of nutraceuticals, increasingly claimed as an imperative to wellness and as an alternative to modern medicines. In recent years many studies have postulated the use of nutraceuticals as therapeutic tools on the basis of their promising effects, with the ultimate goal to validate their efficacy in increasing life expectancy, improving quality of life in elderly and, last but not least, containing the overall cost of health care [33].

The majority of nutraceuticals are phytochemicals or phytonutrients, natural chemicals mainly contained in foods of plant origin (phyto derives from the Greek ϕυτόν and means plant).

These nutrients are not essential for survival, unlike vitamins for example, but are endowed with important beneficial properties for the human body due to their antioxidant, anti-estrogenic, anti-inflammatory, immunomodulatory and anticarcinogenic activities [34]. Moreover, they exhibit a low potency as bioactive compounds when compared to drugs, but since are frequently consumed in substantial amount as part of the diet, can have significant long-term physiological effects [35]. Phytonutrients or phytochemicals can therefore act against chronic inflammations and can be able to help to prevent chronic and aging-related diseases such as arthritis and atherosclerosis.

Over twenty thousand phytonutrients or phytochemicals have been so far identified in plants and, among all of them, flavonoids and carotenoids are the best known.

The above-mentioned bioactive compounds have significant effects on the intestinal environment by modulating the composition of the microbiota. In turn, the biological roles of functional food components can vary because their metabolites and the relative effects can be influenced by the gut microbiota itself and even change from one individual to another [34].

In general, the absorption of phenolic phytochemicals is poor and most of the ingested products reach the colon where they are broken down by the colon microflora to produce different metabolites. The transformation of soy isoflavones into equol and desmethyl angolesin, the transformation of hop isoxanthohumol to make prenylnaringenin (much more estrogenic), the transformation of anthocyanins and procyanidins into phenyl acetic and phenyl propionic metabolites, or the transformation of ellagic acid in urolithins are good examples of the degradation and transformation of polyphenols by the microflora in the colon. These transformations are largely influenced by the nature and characteristics of microflora [36]. Depending on the microorganisms present in the colon, an individual can be a “producer of equol” or a “producer of non-equols”, a “producer of urolithin” or “producer of non-urolithin” and as a consequence, the activity of the phytochemical taken up with the diet can greatly vary [37]. The differences in colon microflora between individuals help to explain the large inter-individual variability and discrepancies in the outcome of clinical trials.

However, when dealing about nutraceuticals it should be kept in mind that the compounds under study refer to the original phytochemicals, i.e., those present in the plant, and not to the relevant metabolites that can be produced during the digestion and adsorption processes in vivo (mainly because these latter are not commercially available). It is as very intuitive that any extrapolation to human of the results obtained in animal models is not appropriate due to their different physiology and to the difference in the bioavailability and metabolism of the active compounds. These gaps, which also occur between different animal models (i.e., between mice and rats), can account for some contradictory experimental results. Currently, although the bioactivity of several functional foods (nutraceuticals) has been extensively investigated by means of both animal and in vitro studies, clinical trials in human are scarce and inconclusive. It must be considered that any chemical alteration of the original bioactive compound, occurring during storage or digestion, may severely modify the bioavailability as well as the bioactivity of the compound itself.

To clarify this concept, it should be consider that even if the antioxidant activity of a compound, as determined in vitro, can be considered a quality indicator of the product, it could not be the real indicator of its ‘goodness’ in vivo, i.e., in the human body. This is the case of pomegranate ellagitannins (ETs). The high antioxidant power of ETs makes pomegranate one of the most powerful in vitro antioxidants [38]. However, in vivo, these compounds are not absorbed and are extensively metabolized by the colon microflora to urolithins which are no more endowed with the original antioxidant capacity of the ETs [39].

Generally, the studies investigating the biological activity of phytochemicals have been carried out using in vitro human cultured cells and (or) animal models; thus, the concentrations tested are often unrealistic with respect to the in vivo situation. Moreover, as already stated, the assayed compounds are the phytochemicals originally present in the plant and not the relevant metabolites produced in vivo (see the above pomegranate ellagitannins example). Thus, it is not surprising that the results of the clinical studies carried out with both healthy and unhealthy volunteers, often show lack of consistency with a large inter-individual variability. In addition, difference in the chemical composition of the nutraceuticals under study and in the pharmaceutical form of choice (pills, capsules, gels, etc.), which can influence the stability and bioavailability of the compounds; also, the physiological status of the volunteers needs attention. The conversion of phytochemicals into metabolites is largely affected by the nature and characteristics of the colon microflora. Thus, it follows that the colon microflora on its own creates differences among individuals and can be responsible for the large inter-individual variability and discrepancies so far observed in the outcome of clinical assays.

The purpose of this article is to clarify the state-of-the-art and to provide current knowledge on the action and clinical benefit of some nutraceuticals in AD.

It is known that onion, garlic, grapes, rosemary, broccoli, spinach, turmeric, parsley, etc., possess considerable antioxidant activities [38,40] and prevent several neurodegenerative diseases including AD [41]. For example, the odorless aged garlic extract (AGE) contains antioxidant chemicals that prevent oxidative damage [42]. AGE has been shown to protect against cellular damage that occurs due to the accumulation of β-amyloid peptide (Aβ) in vitro and to counteract the toxicity of Aβ [42]. Furthermore, always in relation to AD pathogenesis, AGE can reduce the hyperphosphorylation of tau protein by modulating the activity of glycogen synthase kinase-3β (GSK) [42]. The neuroprotective effects of AGE on AD-related cognitive dysfunction have also been explored taking advantage of the ability of the active components of garlic to pass through the blood–brain barrier (BBB), a property which further supports its potential as a therapeutic compound. On the other hand, the use of bioactive compounds such as curcumin, that has beneficial effects on the treatment of Alzheimer’s [27], is limited due to its low solubility and poor oral bioavailability [27]. In other words, the different ability of a bioactive component to cross or not the BBB can make the difference so much so that this feature is now worthy of consideration in the AD treatment approach.

### 2.1. Classification of Nutraceuticals

Considering the significant increase in the consumption of nutraceuticals, witnessed in the last decade, it can be deduced that nutraceuticals per se generate a great expectation in consumers notwithstanding the lack of clear scientific evidence and explanation.

Now, before tackling the cumbersome AD neurodegenerative problem, it is needed to better clarify what nutraceuticals are and how they are classified.

The definitions of nutraceuticals, accepted by the Scientific Societies, are almost these: (i) food or part of it with pharmaceutical properties and with beneficial effects on health, which can be used for preventive and/or curative purposes [43]; (ii) dietary supplements that release high concentrations of biologically active substances of a food nature that have beneficial effects on health [44]; (iii) foods, vitamins, minerals or plant substances that can be ingested in the form of tablets, capsules or drinks providing health benefits [45]; and (iv) dietary supplements that provide health benefit in addition to the standard diet [45].

First of all, it should be emphasized that there is a bylaw-issued by the European Community-which establishes that the beneficial effect of nutrients on health and/or on the reduction of the risk of disease must be scientifically documented (art.6 of the European Parliament’s bylaw concerning nutritional and health claims provided on food products of 20/12/2006 N. 1924/2006).

In the same legislation, the definition of nutraceuticals is not included, therefore nutraceuticals, according to current legislation, are counted as food supplements, without any explanation about the beneficial effects of nutraceuticals on health. Indeed, there are substantial differences (see Figure 1) between nutraceuticals and food supplements [46]. Nutraceuticals, containing active principles of plant and/or animal origin concentrated in pharmaceutical preparations (tablets, capsules, solutions), are endowed with beneficial properties for the state of health. These beneficial properties are supported by scientific data and clinical studies. They contribute to the prevention and/or treatment of diseases. On the other hand, food supplements, in general represented by vitamins, salts, amino acids, plant extracts, etc., are present in the normal diet without any prevention and/or cure ability; they are given as supplement in case of food shortages.

There is therefore no doubt that nutraceuticals, as compared to food supplements, have peculiar characteristics closer to drugs. Consistently, they have been placed as intervention to be implemented “beyond the diet and before the drugs”, in order to indicate their value as a tool in disease prevention [46].

Another distinction should be made between nutraceuticals and functional foods: while the first indicate specific substances extracted from foods that are endowed with medicinal properties, the second refers to the whole food consisting of both the nutritional and the beneficial components. However, the terms nutraceutical and functional food are often used interchangeably as synonyms; in particular, when functional foods, besides to provide nutriment, also help to prevent and treat the disease [47] as in the case of EVOO (see below).

Commonly, nutraceuticals can be grouped into the three main categories including Nutrients, Herbals and Food supplements [48].

To Nutrients belong substances with established nutritional functions, such as vitamins, minerals, amino acids and fatty acids, that are naturally present in our diet and many potential benefits have been attributed to them. Most vegetables, whole grains, dairy products, fruits and animal products such as meat, poultry, contain vitamins and are useful in the treatment of heart disease, stroke, cataracts, osteoporosis, diabetes and cancer. The minerals present in plants, animals and dairy products are useful in osteoporosis and anemia, to build bones, teeth and strong muscles, to improve nerve impulses and heart rhythm.

In the group of Herbals converge herbs or botanical products, as concentrates and extracts. Old as human civilization, garlic, wheat grass, aloe vera and ginger are only some of the natural plant sources that are commonly used in modern nutrition. Among others, there is willow bark (*Salix nigra*), characterized by an active component, i.e., salicin, with anti-inflammatory, analgesic, antipyretic, astringent and anti-arthritic activity. Parsley (petroselinum cripsum) contains flavonoids and possesses diuretic, carminative and antipyretic properties. Peppermint (Mentha piperita) contains menthol, an active component to treat colds and flu, and lavender (Lavandula angustifolia), which contains tannin, is useful for treating depression.

The third group of Food supplements concern a set of reagents or a mixture of them, orally administered, that are used as nutritional support in specific situations as sports, weight-loss and meal replacements. They contain a dietary ingredient intended to add something to the foods that can improve health status. Examples of dietary supplements are black cohosh for menopausal symptoms, ginkgo biloba for memory loss and glucosamine/chondroitin for arthritis.

Far from wanting to be repetitive, compared to colleagues who preceded us in dealing with this type of study, we believe that giving general indications of nutraceutical classification, in this context, could be worthwhile.

The term nutraceutical refers to a broad class of compounds that includes many categories and sub-categories. Below, a rapid classification of nutraceuticals on the basis of the (i) origin source: vegetable, animal and microbial; (ii) mechanism of action and (iii) chemistry nature is reported, according to [49,50,51].
(i)Classification of nutraceuticals based on origin: vegetable, animal and microbial.Not only foods from plant or animal origin are sources of nutraceuticals, but also bacteria. In some cases, the identification of the source is not immediate; for example, conjugated linoleic acid, which is essential to human nutrition, is mainly found in animal food such as beef and dairy products, but only because it is produced by the bacteria present in the rumen of the cows. Many nutraceuticals have a high conserved biochemical structure between species and therefore can be found in both animal and vegetable or can be a microbe’s by-product. This applies to choline and phosphatidylcholine, for example.(ii)Classification of nutraceuticals based on the mechanism of actionThe classification according to the mechanism of action is widely used by doctors, nutritionists, dietitians and, generally speaking, by those who provide people with useful indications to improve their health. Hence, who deals with cardiovascular diseases will be more interested in those compounds improving the lipid profile and with anti-inflammatory action, as well as who deals with oncology will be more interested in knowing those compounds having anticancer properties.(iii)Classification of nutraceuticals based on their chemistry natureBased on the chemical nature of the different compounds, consistently with [52], nutraceuticals can be classified as:(1)Dietary fibers: substances of vegetable origin present in foods that are not metabolized in the large digestive tract and increase volume of the intestinal content. Chemically, dietary fiber means carbohydrate polymers with a degree of polymerization not lower than 3, which are neither digested nor absorbed in the small intestine. Examples include fruit, barley, oats, lignin, cellulose, pectin, etc. Generous intake of these fibers by diet is associated with a low risk of cardiovascular disease, hypertension, diabetes, obesity, colon cancer and gastrointestinal disorders.(2)Probiotics: probiotics are food components consisting of live microbes that have many beneficial effects on the human body. Diets rich in probiotics, prepared naturally or by industrial fermentation processes, have shown positive effects due to improvement of the intestinal microbiota.(3)Prebiotics: unlike probiotics, they are not live organisms. They are short-chain polysaccharides that have unique chemical structures-in particular fructose-based oligosaccharides that exist naturally in food or are added in the food-that are not digested by humans, but still have many useful effects on our body. In the colon, the structure and performance of the microbiota change, thus influencing the growth and action of specific bacteria improving the health of the host. Prebiotics represent the nourishment of probiotics and stimulate their activity in the gastro-intestinal tract. Examples of these foods are the roots of chicory, banana, tomato, allium, beans, etc.(4)Polyunsaturated fatty acids: omega 3 fatty acids, e.g., α-linolenic acid, eicosapentaenoic acid and docosahexaenoic acid, are present in oily fish, flax seeds, soybeans, etc. or omega 6 fatty acids, e.g., α-linoleic acid and arachidonic acid, are found in corn, safflower, sunflower and soybean, etc.(5)Antioxidant vitamins: vitamin C, vitamin E and carotenoids. These vitamins are abundant in many fruits and vegetables. Regular intake of them helps to prevent a range of diseases.(6)Polyphenols: they are phytochemicals produced by plants to protect against photosynthetic stress and reactive oxygen species. Among the others, flavonoids, anthocyanins and phenolic acids, found in a variety of foods, have anti-inflammatory and antioxidant properties.(7)Spices: esoteric food additives used to improve the sensory quality of food. Most of the spices are terpenes and other components of essential oils. Minimal amount of diet spices has antioxidant, chemopreventive, anti-mutagenic, anti-inflammatory and immune effects.

Moreover, some nutraceuticals are present in more than one food, e.g., quercetin is present in onions, as well as in broccoli or red grapes. For this reason, when referring to a substance or a family of nutraceutical substances, it is worthwhile to mention the food sources richer in it. Even the soils on which the plants grow or the type of feeding of the animals affect the quantity of nutraceuticals present in food, so much so that it is not the same thing to eat a beautiful red tomato ripened in the sun or only a rosé tomato.

A further classification of nutraceuticals, that sometimes is used, concerns the ‘promise’ of their pro-health activity. Therefore, it is possible to deal with “potential” nutraceuticals and “consolidated” nutraceuticals. A potential nutraceutical is one that promises particular benefits; this nutraceutical becomes consolidated only after sufficient clinical data have been collected to demonstrate his benefit. Unfortunately, the majority of nutraceutical products belong to the “potential” category and are still waiting for validation [53].

Further, although nutraceuticals were advertised with the claim to be safe and effective only because they are either a food or a natural food derivative [27], unfortunately, nutraceuticals have also several drawbacks [54]. In fact, side effects and toxicity are continuously reported not only because of the ingestion of nutraceutical itself, but also because of the possibility of contamination by pesticides, other toxic plants, metals, fertilizers, etc. [27].

The potential health benefits of nutraceuticals together with the disadvantages and the side effects that, sometimes confirm their potential toxicity are reported in Table 1.

Other drawbacks—that have led researchers to discover and develop better methods and new strategies to optimize the bioavailability, pharmacokinetics and efficacy of these products—include ineffective targeting, low solubility, rapid degradation in the intestine, poor permeability (e.g., through BBB), fast metabolism, short half-life and others [27,55].

Further limitations may be caused by the fact that the bioactivity of a native herbal product could be modified by metabolic reactions in vivo and/or by interactions with the intestinal microbiota, as already mentioned [56]. Advances in knowledge about the interactions between bioactive food compounds and specific intestinal bacteria could contribute to a better understanding of the in vivo effects of nutraceuticals.

## 3. Alzheimer: Molecular Aspects and Causes of the Disease

Brain cells make us more or less what we are and allow us to carry out all the activities of life. Therefore, it is easy to understand how any damage to these cells can affect a wide range of the daily life processes [57], including:remember (both short and long term) and recognize,speak and write,make decisions and solve problems,interpret sensory input,orientate and juggle yourself the world.

With enough damage to neurons (Figure 2), cells die.

Studies on AD have been feverish for years. The diagnosis of the disease is first based on the evaluation of neuropsychological and behavioral symptoms related to disturbances of memory, language and visual–spatial perception.

The cause of this disease is generally assumed to be the interaction between environmental factors and genetic components in most cases [58]. The latest by themselves are responsible for about 10% of the AD cases in general population but for about 100% of cases when the disease occurs before age 40-50. This early-onset AD form is caused by mutations in three genes that encode specific proteins, i.e., β-Amyloid precursor protein (APP) on chromosome 21, presenilin-1 (PSEN1) on chromosome 14 and presenilin-2 (PSEN2) on chromosome 1 with the latter covering alone about 1% of cases [59].

Other genes, when mutated, represent a cause of greater susceptibility for the onset of the late form of the disease. Among them, there are some variants of the APOE gene on chromosome 19, such as the APOE4 gene, which encodes a protein playing an important function in cholesterol metabolism. However, unlike APP and presenilins this gene represents only a risk factor for the late-onset AD in 60% of cases [60,61,62]. In other words, its presence per se does not represent an illness cause.

Thus, as far as genetics involvement in the pathogenesis of AD is concerned, we know the molecular nature of the single gene mutations, but we still lack the intimate mechanism by which these mutations cause the anatomopathological changes found by Alois Alzheimer.

### 3.1. β-Amyloid and Tau Proteins

As first observed over 100 years ago, the most distinctive diagnostic traits of AD are the so-called amyloid plaques, i.e., accumulations of β-amyloid protein outside neurons, and, inside the cells, tangles of abnormally modified Tau protein: both phenomena are attributed to neuronal damage occurring in the course of pathology, but not always.

Together related with the loss of memory and the destructuring of cognitive abilities, determined by deterioration of the synapses, both proteins, Aβ and Tau, represent a characteristic hallmark of AD (Figure 2). They begin to accumulate in the brain several years before clinical symptoms appear. Age-related plaques can be found in specific brain areas such as the hippocampus, amygdala and neocortex [63,64]. The Aβ peptide is a small peptide derived from the proteolytic cleavage of APP by β-secretase and γ-secretase along the secretory amyloidogenic pathway taking place in many neuronal compartments, including axons, nerve terminals and dendrites [65,66]. APP is a protein located in the neuronal membrane of nerve cell; its function is not yet known but it has been proposed to be involved in neuronal differentiation as well as in the growth of nerve fibers. In cell, when APP no longer functions properly, it undergoes the degradation pathways common to all cellular proteins. If, for unknown reasons, the APP is abnormally or excessively degraded, the resulting β-amyloid peptide is secreted outside the cell and begins its destructive process by aggregating into small clusters that alter the neuronal communication in AD brain. The most abundant species of Aβ is the Aβ40 fragment. The Aβ42 isoform is less abundant, but more prone to form amyloid fibrils, which accumulate in the brain of AD patients [67,68]. The increasing accumulation of extracellular Aβ destroys synapses and so the massacre continues in an infinite loop that could partly explain why many drug trials targeting this form of dementia do not work.

These events take place long before the appearance of the characteristic plaques, as well as the deposition of Aβ begins long before the development of clinical dementia and can be ante-mortem revealed by PET-amyloid imaging in the brain of AD subjects [69,70,71]. It is important to note that brain autopsy of non-demented persons often show extensive amyloid plaques as well as PET-amyloid imaging indicates diffuse deposition of Aβ oligomers occurring also in the brain of healthy aged individuals without AD [69,72]. It is widely accepted that, unlike the insoluble extracellular senile plaques made of aggregated Aβ peptide, the soluble pre-fibrillar Aβ oligomers (Aβos) are the most neurotoxic species by playing a critical role in the early synaptic dysfunction associated with the pathogenesis of AD. Aβos represent a heterogeneous population of aggregation states including dimers, trimers, Aβ*56 and spherical oligomers which differ in size, morphology and cytotoxicity. They operate at different stages of the clinical progression of AD, through multiple neurotoxic mechanisms, such as receptor binding and disturbances of receptor-activated signal transduction pathway, mitochondrial dysfunction, dysregulation of Ca^2+^ homeostasis, alterations of tau metabolism [73].

Another protein claimed to be causative of AD is tau, a key component of microtubules, the cell structures contributing to neuronal stability. Tau is a microtubule-associated protein characterized by a complex biology, which includes multiple splicing variants and several phosphorylation sites. Tau phosphorylation is part of the normal microtubule assembly process, but, in AD, tau becomes hyperphosphorylated or glycosylated, thus weakening its affinity for microtubules and favoring its aggregation into filaments which accumulate inside the cell with formation of the neurofibrillary tangles (NFT). Accumulating over time, tangles promote cytoskeleton degeneration and neuronal death and then, starting from memory loss up to the involvement of all the other cognitive functions, symptoms of the disease develop progressively. Currently, potential compounds that act against the accumulation of hyperphosphorylated tau protein are considered potential therapeutic agents, including inhibitors of the kinase that promote tau phosphorylation [74,75].

About the crucial question regarding the two potentially “dangerous” proteins to which the responsibility of giving first the green light to the disease is attributed, in light of the recent data [76,77,78] it is clear that Aβ and tau pathologies exert cooperative and/or synergistic deleterious effects on neuronal function(s) (“dual hit” hypothesis), particularly at synapses which are believed to initiate the AD progression.

What is certain is that to date, it has been excluded that Aβ and tau alterations are the only cause of dementia, otherwise it would not explain why about half the population, at any latitude, after 45 years of age has plaques and tangles without becoming demented. As well as amyloid plaques, without causing any disturbances, have also been found in the brain of superagers, a word that identifies the elderly with brilliant cognitive functions. However, being first recognized in the brains of the demented people, the plaques and tangles were blamed for the disaster, together with other markers of neuronal damage, such as oxidative damage to DNA, lipid and carbohydrate and redox proteomics data show that the degree of oxidative damage of specific proteins is more severe that in non-pathological aged brain [79,80,81].

After a huge research, in 2018 three scientists came to four main conclusions [82]:In the brain without cognitive impairment there can be Aβ and tau deposition.Clinical diagnosis of AD does not involve the status of Aβ and tau in the brain.Size and enlargement of the plaques are not related to cognitive impairment.Amyloid in the brain is not a warning sign of dementia.

In most cases, AD is not due to a single causative event, but it is a complex syndrome due to several causes. Therefore, if treated in time or prevented, the number of demented subjects should drop, without completely disappearing due to the persistence of the AD form caused by genetic mutations. In 2017, the World Health Organization placed preventive measures at the top of the priorities to combat dementia [83,84]. Timely prevention and treatment of other diseases and/or disorders and/or modus vivendi—e.g., arterial hypertension and other cardiovascular disorders, diabetes, obesity, depression, smoking, alcoholism, poor mental commitment, physical inactivity, insomnia, feeding [85]—lower the risk of dementia by up to 50%. For example, from the age of 45, blood pressure can begin to rise without apparent disturbance, until brain damage occurs over time. Therefore, after 45, blood pressure control should be a daily rule to set up [86].

### 3.2. Apoptosis

A unifying hypothesis identifies the triggering event of AD in apoptosis (Figure 2), an event affecting large neuronal populations. In turn, one of the causes that could provoke an abnormal activation of the self-elimination program for apoptosis of entire neuronal districts would concern neurotrophins. This family of molecules, which has the Nerve growth factor (NGF) as its progenitor, mainly performs the function of keeping the neuronal death program blocked: therefore, a neuronal population deprived of the basal supply of specific neurotrophins undergoes massive apoptosis [87,88,89]. It occurs not only in cell cultures but also in the transgenic animal models lacking NGF which develop an AD-like syndrome with comparable anatomopathological and behavioral features.

Another cause that can activate the apoptotic program in cell cultures or laboratory animals is the deprivation of electrical stimuli. When electrical stimuli are interrupted due to a traumatic rupture of the afferent fibers, a massive apoptosis program is activated. Compelling evidence indicates that excessive potassium (K^+^) efflux and intracellular K^+^ depletion are the key early steps in apoptosis. The results obtained in in vitro cell cultures or in animal models unequivocally confirm that there is a causal relationship between the events activating apoptosis and AD [90,91].

As a consequence of the improper activation of the apoptotic program, in some neuronal populations, an altered degradation of APP and tau would occur with consequent triggering of the death cascade in neighboring cells, cells that are still healthy and that otherwise would not have died.

The widespread death of many subtypes of neurons, observed in AD, begins in the hippocampus and progressively spreads in the cortex [92,93]. However, this death varies in its features, showing in various proportions the characteristic of either apoptosis or necrosis, the two historically recognized forms of cell death. Furthermore, recently emerging evidence has discovered a new genetically programmed and regulated form of necrosis called necroptosis. Initially identified as a by-product of inflammatory stimuli, it has been subsequently ascertained that necroptosis: it is induced by the activation of specific ligands under the presence of numerous factors, both intra- and extra-cellular. The necroptotic signaling path leads to the formation of a multiprotein complex, the “necrosome”, whose activation induces various biochemical and structural modifications leading to cell death. Necroptosis also appears to play a key role in AD [94,95,96].

### 3.3. Mitochondrial Dysfunction and Oxidative Stress

Defects in energy metabolism are a consistent feature of the brain affected by AD. In fact, there is a close association between the onset of cognitive impairment and the impairment of energy production in the brain; the appearance of metabolic abnormalities also seems to precede the clinical onset of the disease by about ten years [97]. Extensive research reports that at least two precocious events are distinctive of the disease: (i) mitochondria dysfunction, responsible for the reduction of the energy metabolism in the brain and (ii) increase in oxidative stress.

Studies conducted on post-mortem brain tissue have shown that a reduction in the activity of some enzymes of the tricarboxylic acid cycle is observed in Alzheimer’s patients [98,99,100]. Bubber and collaborators [101], carrying out a complete analysis of the activity of the entire enzyme machinery of the Krebs cycle in AD brain tissue, confirmed the decrease in the activity of pyruvate dehydrogenase and αketoglutarate dehydrogenase complexes, together with a significant reduction in the isocitrate dehydrogenase activity; instead, the succinate dehydrogenase and malate dehydrogenase enzymes showed an increased activity. All the others are unchanged. The low COX activity has been observed by many authors in hippocampus, temporal cortex and at peripheral tissue level in platelets and fibroblasts of patients with sporadic forms of AD [102,103,104]. Further, we found that Cytochrome c (cyt c) is released from the mitochondria while still coupled [105] and an increase in the ATP level [105] as well as an alteration of the adenine nucleotide translocator-1 (ANT-1) occur, with ANT-1 becoming a component of the mitochondrial permeability transition pore [105].

The involvement of oxidative phosphorylation (OXPHOS) complexes appears more controversial, however the functional anomalies of the Krebs cycle and of the respiratory chain inevitably induce an alteration of the energy metabolism and lead to an increased production of reactive oxygen species (ROS): exceeding a certain critical threshold contributes to modify the permeability of the mitochondrial membrane that, like a switch, triggers the cell death programmed which underlies the neurodegenerative process [101,105]

Therefore, in addition to serving as subcellular organelles, essential to generate the energy that powers normal cell functions, mitochondria also monitor cell health and, if necessary, initiate cell apoptosis.

Mitochondrial dysfunction is probably one of the earliest events that occur in the course of AD. Damaged mitochondria would become less efficient producers of ATP and more efficient producers of free radicals [105,106], two processes which lead to a state of chronic oxidative stress when antioxidant mechanisms are no longer able to keep up with the production of ROS.

It seems clear that oxidative stress is one of the triggering factors.

Oxidative damage is known to occur long before the formation of the Aβ plaques, and this qualifies both mitochondrial dysfunction and oxidative stress in AD as early actors in the pathological process. Autoptic studies on brain tissue of AD patients have confirmed the presence of numerous signs of oxidative stress such as an increase in lipid peroxidation, oxidation of proteins and glycides and reduction in antioxidant enzymes [105,107]. Furthermore, in vitro studies have shown that the neurotoxic properties of Aβ can be mediated by ROS [108,109].

The action of Aβ and APP on mitochondria would explain the abnormal amount of free radicals and oxidative stress, the impairment of the activity of the enzymes of the intermediate metabolism and the mitochondrial dysfunction observed both in the disease and in the cells intoxicated with the toxic peptide [110,111,112,113,114]

Since the accumulation of Aβ causes further ROS production, it is still not clear whether excessive oxidative stress is a primary or secondary event in AD [115]. However, this aspect appears to be of relative importance, as ROS production, even if only secondary, is extremely harmful to brain tissue because beside contribute to neuronal damage, one of the most serious free radical attack on the body is memory loss, therefore any effort aimed at removing ROS or at preventing ROS formation may be of primary use in AD patients. It is also known that oxidative stress favors Aβ production by stimulating the activity of Beta-Site APP-Cleaving Enzyme (BACE). Plaque formation occurs only when APP protein comes into contact with BACE, which cleavages APP allowing it to precipitate in the form of plaques [116,117]. All these data constitute an element in favor of the use of antioxidants for the prophylaxis and treatment of the disease.

However, there is not clear evidence on their clinical efficacy.

### 3.4. Microbiota and Diet

The role of intestine in the development of neurodegenerative diseases including AD is currently under intense investigation. A large body of research is investigating the mechanism(s) of how

i.microbiota cooperates with the host to maintain optimal health status of the individual;ii.nutrition influences the intestinal microbiota and the risk to develop AD.

Then, once again, the role of diet in the development and prevention of disease is in the spotlight. Nutrients have been shown to modify the composition of the gut microbiota and, consequently, the formation of Aβ plaques.

The relationship is now certain: an inflamed intestine causes inflammation of the brain [118].

If microbiota, the population of billions of bacteria living in our intestine and that are likely to have an effect on brain functions, is in condition of dysbiosis, this causes an inflammatory stress not only in the gut, but also in the brain. Numerous tests suggest that the intestinal microbiota plays an important role in brain development and that there is a two-way relationship, between brain, intestines and bacteria colonizing the intestine, which has been identified as the brain–intestine–microbiome axis, able to modulate the functions of the gut, the immune as well as the central nervous system (CNS) [119,120].

The existence of a cross-talk between the gut and brain is already known. The nervous circuits involved in eating behaviors are precisely coordinated by the brain centers that regulate energy homeostasis and cognitive function. Various studies, conducted on animals and humans, demonstrate how the formation of synapses requires the presence of precursors and nutritional cofactors.

It has been shown that microbiota communicates with the brain through the activation of specific complement fragments (C1q) and pro-inflammatory signal receptors that could affect some features of AD, such as the production and deposition of amyloid plaques [121,122].

Moreover, recent discoveries show that the abundance of particular bacterial species in the intestine (as well as in the oral cavity) could influence the Aβ deposition and lead to the development of AD.

The first evidence supporting the link between intestinal microbiota and AD derives from experimental mouse models. In particular, it has been shown that the mnesic abilities of mice without bacterial colonization are strongly compromised compared to mice with an intact intestinal microbiota. In addition, the use of probiotics has been able to prevent the mnesic deficits induced by gastroenteritis.

Gut microbiota dysbiosis in patients with AD could trigger amyloid inflammation-induced formation and aggregation, on the contrary maintaining a health gut microbiota is increasingly proving to be an effective strategy for preventing or reducing the risk of AD [123,124]. However, still no evidence exists about a strong prevalence of pro-inflammatory bacteria at the expenses of the anti-inflammatory ones in AD patients.

It also appears that the AD pathological markers in cerebrospinal fluid for AD, such as the levels of Aβ protein, could correlated with a change in the number of “good” bacteria in the intestine.

For example, species belonging to the Bifidobacteriaceae, Ruminococcaceae, Peptostreptococcaceae, Turicibacteraceae, Mogibacteriaceae, Clostridiaceae families are less abundant in Alzheimer’s patients, while Bacteroidaceae, Gemellaceae and Rikenellaceae families are more abundant [125,126,127].

A decrease in the number of *Bifidobacterium* could directly be correlated with the high levels of Aβ in the cerebro-spinal fluid of Alzheimer’s patients.

Preliminary clinical investigations conducted on several AD patients indicate that diet integration with antioxidant and anti-inflammatory nutrients is able to promote the growth of *Bifidobatcterium*, which is necessary for the microbiota eubiosis (maintaining the balance of the microbiota) and therefore capable of preventing the production of bacterial toxins involved in brain amyloidogenesis (formation and aggregation of β-amyloid), thus effectively improving the cognitive ability of patients [128].

Therefore, if the genes inherited from the parents are difficult to change, this is not the case of intestinal microbiota which can be modulated by diet and dietary supplements.

Extensive experimental studies in animals, together with many prospective and cross-sectional epidemiological studies performed in humans, have already strongly suggested the beneficial role of nutraceuticals and similar compounds, as well as the microbiota involvement, in preventing memory and cognitive loss, proper of AD [8].

The modulation of gut microbiota could be achieved by adopting and maintaining a healthy diet and may be a winning strategy in the prevention of AD.

Taking into account that the first symptoms tend to appear in old age compared to the silent onset of the disease, the research effort has moved towards preventive methods. Surely, this type of preventive intervention-modulating lifestyle and diet-can be too late in people aged 75 years and over [129], but it could be fruitful in younger population, especially from dementia risk groups, who have not yet developed the disease. Available information strongly supports the notion that apart from preventing the onset of neuronal damage, nutraceuticals can potentially attenuate the continued progression of neuronal destruction.

This review report will be centered only on some natural products for which mechanistic evidence for neuroprotection are available, e.g., genistein, curcumin, RSV, polyphenols present in extra virgin olive oil, and red wine and others. We will describe how they present in a wide variety of plants, fruits and seeds and how they, if taken with the diet, can improve the damage associated with the cognitive decline of Alzheimer brain by correcting biochemical alterations at multiple levels. In addition, it will also be considered not only the state of the microbiota, as a potential disrupting variable in the interaction of major bioactive compounds, both protective and detrimental, but space will also be given to studies concerning nutrient-dependent modifications of the epigenome which, as known, affect the progression of the disease.

## 4. Nutraceuticals for Neuroprotection and Treatment of Alzheimer’s Disease

The loss of the ability to mind life remembrances is a hallmark of the early AD.

Several years of intense research have revealed that AD is related to numerous cellular changes, including biochemical alterations, such as mitochondrial damage and oxidative stress, as well as neuropathological formations, i.e., Aβ synthesis and accumulation, phosphorylation of tau and NFTs formation, loss of synapses and finally loss of neurons [130].

The poor efficacy of acetylcholinesterase inhibitors in stopping or reversing AD progression supports the idea that only early diagnosis and treatment can really safeguard cognitive functions and ensure neuronal survival. It has been widely accepted that the onset of the disease occurs many years before symptoms appear [131]. Considering that multiple factors—genetic, environmental, dietary, or a combination of them—are considered initiators of disease, it is assumed that a certain threshold need to be surpassed before actual disease manifestation arises.

That is why the focus of the research is essentially prevention.

The polyphenols, flanked by the colored carotenoids and other bioactive compounds, essentially belonging to plant kingdom, represent that category of substances defined as “functional”, because they can contribute, with different modes of action, to the well-being of the organism [132]. They are extensively drawing attention as potential tools both to prevent AD and to slow down its course [132].

It was not possible to browse all possible supplements that can be used in this context. Therefore, precise choices were made in the selection of nutraceuticals in terms of anti-Alzheimer’s medicine, opting for the most consolidated ones in terms of the medical-scientific reference literature, most frequently used in clinical practice and endowed with an action as possible multifaceted.

Surely, many substances, as such or as valuable precursors of compounds, are essential to neuronal metabolism and brain health maintenance. Some of them deserved to be at least mentioned both for their fundamental contribution to the right functioning of the CNS and for the deleterious repercussions on cognitiveness when they are lacking. We refer for example to mono- (MUFA) and polyunsaturated fatty acids (PUFA, ω-3 and ω-6), whose main food sources are fish and vegetable oils or also to the B vitamins, folic acid, vitamins B6 and B12, essential substances for normal neuronal functioning and well known for decades for their beneficial effects on cognitive and behavioral performances [133,134]. However, we preferred to turn our study to nutraceuticals, specifically studied for their nootropic activity, reason why they are called ‘smart drugs’ and/or ‘smart nutrients’.

The nutraceutical scenario mainly places on the podium the polyphenols, natural antioxidants, with their beneficial effects on health and their ability to contrast or slowdown degenerative diseases [32]. Polyphenols are the most abundant antioxidants in our diet and are commonly present in fruits, vegetables, cereals, olives, dry legumes, licorice, chocolate and beverages, such as tea, coffee and wine [28,135].

Polyphenols, mainly flavonoids, are specific of particular foods, such as flavanones in citrus fruits [136], isoflavones in soy [137,138], but other polyphenols, such as flavan-3-ols (sometimes referred to as flavanols), one of the most prominent groups of flavonoids for their beneficial properties [137] which include catechin, epicatechin, epigallocatechin, epigallocatechin gallate, are found in a plethora of vegetable products, as cocoa, chocolate, as well as in black and green tea and grapes. In addition to nutraceutical and functional foods containing innumerable bioactive components, noteworthy is EVOO, a ‘must’ of the MD.

Here, we will deeply describe only some polyphenols and/or carotenoids used as nutritional supplements summarizing the current knowledge about them, spanning from dietary sources to their neuroprotective capabilities. Nutraceuticals will be organized into four groups according to their protective action against:oxidative stress and mitochondrial dysfunction,Aβ and tau toxicity and aggregation,neuronal damage and apoptosis,memory loss and cognitive decline.

### 4.1. Nutraceutical Compounds against Oxidative Stress and Mitochondrial Dysfunction

Mitochondrial dysfunction and oxidative stress, leading to neural membrane injury and memory impairment [110,138,139,140,141,142,143] are some of the biochemical features occurring in AD, as widely discussed in Section 3.3.

Mitochondria, damaged during oxidative stress, in turn start to produce ROS increasing the production of Aβ peptides, which further induces oxidative stress both in vitro and in vivo [140], leading to neuronal apoptosis and to an acceleration of AD progression [140].

#### 4.1.1. Genistein

Genistein (GEN), an isoflavone mainly found in soy products, is a potentially effective compound in preventing AD [144]. Owing to the presence of numerous phenolic moieties in its structure, GEN exhibits potent antioxidant property [145]. Therefore, it is not surprising that GEN has antioxidant and neuroprotective effects against in vitro chemically induced AD [146]. It reduces oxidative stress by inhibiting the synthesis of ROS; it also protects mitochondria by increasing the reduced to oxidized glutathione ratio and lowering 8-oxo-20-deoxyguanosine, a marker of mitochondrial DNA damage [147].

In a study conducted on primary cultured neurons, its neuroprotective effect has been verified on cells treated with the Aβ fragment 31–35 (Aβ31–35).

A lot of literature has reported that the short fragment Aβ31–35, obtained by proteolytic cleavage of Aβ precursor protein (APP), could cause apoptosis in primary cultured neurons [148] and PC12 cells [149], acting as an ‘active center’ of the full length Aβ molecule with stronger neuron toxicity [148,150]. Ding et al. [151] observed that GEN addition to neurons, 2 h before challenge with the Aβ 31-35 fragment, was able to rescue neurons by increasing cell viability and reducing both Ca2+ and ROS generation Furthermore, the ratio of GSH⁄GSSG in mitochondria and mitochondrial membrane potential were also increased after GEN treatment.

In addition, in an AD cell model, such a cerebellar granule cells (CGCs) undergoing apoptosis due to potassium deprivation, Atlante et al. [152] observed that GEN was able to prevent apoptosis by reducing impairment of both aerobic glucose metabolism and mitochondrial uncoupling, thus suggesting that the therapeutical target of GEN could be mitochondria. Consistently, in the same study, ROS-dependent cyt c release, ANT alteration and mitochondrial permeability transition pore (mPTP) opening were prevented when GEN was added to cells undergoing apoptosis, thus concluding that the prevention of apoptosis is essentially due to the antioxidant properties of this flavonoid, in particular against hydroxyl radicals. Indeed, GEN has been recognized as a more effective compound in inhibiting oxidation induced by hydroxyl radicals than by superoxide radicals [153].

To date, the effect of GEN, used as dietary supplements on AD patients, has not been tested.

#### 4.1.2. Resveratrol (RSV)

By virtue of the presence of OH groups in position 3, 4 and 5, aromatic rings and a double bond in the molecule, RSV has a strong ability to remove free radicals (Figure 3).

RSV is a phytoalexin that mainly occurs in grapevine species (*Vitis* sp.), but also in other fruits. In particular, the concentration of RSV in red wine is higher than the white one, due to the winemaking process which leads to a more extensive contact between peel/seeds and must [154].

That RSV is found in several berries and related products, such as bilberries, blueberries, cranberries, lingonberries, partridgeberries, mulberries and strawberries, which has been proved by various researchers [155,156,157,158]. However, both technological and agronomic processes applied to berries and the low content of RSV are limiting factors for considering berries as a significant source for this bioactive compound. Similarly, other foods contain small quantities of RSV, such as peanuts, dark cocoa, beer and beeswax from honeycomb [159].

Recently many studies are focusing on RSV, as molecule more in view in the field of the lifestyle-related diseases, including AD [160]. This interest is mainly due to its beneficial effect against oxidative stress and neuroinflammation, by modulating proteins that are closely related to neurological disorders, i.e., AMPK, sirtuin-1 pathway (SIRT1) and receptor-gamma coactivator 1α (PGC-1α) [140,160] (Figure 3).

In vitro studies on human neural stem cells and PC12 cells have revealed that the addition of RSV to the incubation medium increased Glutathione peroxidase, SOD1 and catalase activities, decreased the lipid peroxidation product, malondialdehyde (MDA) and restored GSH levels [140].

The key neuroprotective mechanism of RSV in AD appears to be related to the activation of SIRT1 [160]. SIRT1 regulates the activity of several substrates, including p53 and the peroxisome proliferator-activated PGC-1α [161], a substrate able to improve mitochondrial dysfunction [162]. Moreover, activation of SIRT1 leads to inhibition of nitric oxide synthase (iNOS), which becomes active in response to cell damage and NF-κB signaling. The iNOS inhibition occurs in a double way: via a direct inhibition by SIRT1 or through SIRT1 related activation of suppressor of cytokine signaling 1 (SOCS1) [160]. Both ways lead to a decrease in iNOS expression, and thus to a drop in the NO synthesis, which in the presence of ROS can create the highly neurotoxic reactive nitrogen species, such as peroxynitrous acid (ONOOH), nitrogen dioxide (•NO_2_) or dinitrogen trioxide (N_2_O_3_) [160] (Figure 3).

As mentioned several times, caution is required in implementation of therapies with natural products, since the patient’s intrinsic aspects, environmental factors and characteristics of the compound under study are key elements for the therapeutic efficacy and success. No less, also factors closely related to the experimental system studied and to the nutraceutical concentrations used may influence the benefic effects of compound.

The protective effect of RSV has been validated in several cell models. It has been found that RSV attenuates the oxidative damage induced by Aβ accumulation in SH-SY5Y cells as well as in a rat model of ibotenic acid-induced AD [163]. In a differentiated lineage from rat pheochromocytoma, used as a cellular model of AD when treated with Aβ1–42 (Aβ1–42) peptide, Wang et al. [164] showed that RSV decreased the mitophagy-mediated mitochondrial damage and attenuated the oxidative stress caused by Aβ1–42 [165].

Another neuroprotective property attributed to RSV is the suppression of ROS formation through the inhibition of prooxidative genes (i.e., nicotinamide adenine dinucleotide phosphate oxidase) [140]. Further, Huang et al. [166] showed that the neuroprotective activity of RSV included the suppression of inducible iNOS, which is involved in Aβ-induced lipid peroxidation and heme oxygenase-1 downregulation, thereby protecting the rats from Aβ-induced neurotoxicity [140].

Overall, all these observations suggest that this compound may be included among the others endowed with potential therapeutic for AD. However, despite the neuroprotective potential of RSV has been demonstrated in numerous in vitro studies, the main limitation currently addressed—as for GEN—is the lack of information from clinical studies that correlates the activation of SIRT1, the reduction of the inflammatory and the relative oxidative state with the improvement of the development and progression of AD.

#### 4.1.3. Curcumin

Among natural compounds exerting antioxidant effects, Curcumin (CUR) is one of the best known [167,168].

CUR, a bright yellow polyphenol found in the rhizome of *Curcuma longa*, is used in food as a spice, especially in traditional Indian, Middle Eastern and Thai cuisine. *Curcuma longa* performs a marked anti-inflammatory and antioxidant action, blocking free circulating radicals and inhibiting the formation of new ones.

CUR is a lipophilic compound, which makes it an efficient scavenger of peroxyl radicals. It can modulate the activity of catalase and SOD as well as inhibit ROS-generating enzymes, such as lipoxygenase/cyclooxygenase and xanthine hydrogenase/oxidase [169].

Oxidative stress has been linked to ageing and cognitive decline [170], and the antioxidant effects of CUR have been of much interest in this respect. The antioxidant mechanisms of CUR are also considered central to explain its neuroprotective effects against Aβ (see below), while related compounds from turmeric rhizome (demethoxycurcumin, bisdemethoxycurcumin and calebin A) are also neuroprotective in vitro [171].

CUR antioxidant mechanisms explain its ability to improve memory in many of these studies, including via reduced oxidative damage through enhancing GSH and decreasing lipid peroxide levels in brain tissue [171].

#### 4.1.4. Carotenoid

Carotenoids, plant-derived pigments present in many fruits and vegetables, are responsible for their yellow, orange and red colors [172]. They are widely distributed in nature and consist of a group of more than 700 naturally occurring pigments [173]. Only about 60 of them are present in a typical human diet such as β-carotene, α-carotene, lycopene, lutein and cryptoxanthin [174].

Carotenoids can also be classified on the basis of their functional groups in: (a) carotenes (e.g., α-carotene, β-carotene and lycopene) which are exclusively hydrocarbons [173], without any functional group [172] and (b) xanthophylls (e.g., lutein, zeaxanthin) containing oxygen as functional group [172].

In all organisms, carotenoids function as antioxidants [174]. They scavenge radicals thanks to the presence of conjugated double bonds which enable these compounds to accept electrons from reactive species thus neutralizing free radicals [172].

On the other hand, they can be divided into provitamin A (e.g., β-carotene, α-carotene and β-cryptoxanthin) and non-provitamin A compounds [173]. Regarding provitamin A, its activity has long been known to be associated to carotenoids [173]. Of the approximately 700 carotenoids found in nature, only about 50 have provitamin A activity [173] and only three are the most important precursors of vitamin A in humans: α-carotene, β-carotene and β-cryptoxanthin [173]. The β-carotene, found in fruits and vegetables, is the most famous nutrient exhibiting pro-vitamin A activity [175].

Notwithstanding the great potential of using carotenoids due to their medicinal properties, the use of animal models for studying carotenoids is limited since most of the animals as well as human beings do not absorb or metabolize carotenoids [173].

#### 4.1.5. Lycopene

Belonging to the carotenoid family, Lycopene (LYC) has the greatest antioxidant activity among all dietary carotenoids [173]; but, unlike most of carotenoids, LYC does not have pro-vitamin A properties. It is present in high concentrations in ripe tomatoes and to a lesser extent in watermelon, apricot, grapes, guava, pink grapefruit and papaya. The LYC content is influenced by the level of ripeness of the tomato [176].

Again by virtue of its antioxidant properties, it has neuroprotective potential; indeed it protects from neurodegenerative diseases such as AD and, more generally, dementias and Parkinson’s disease, especially in combination with other active ingredients, such as lutein, astaxanthin and zeaxanthin [177]. Many studies show that free radical level and amyloid-induced mitochondrial dysfunction lowered [178] after LYC treatment. Recently, a study demonstrated that 8 weeks of LYC treatment reversed MDA increase and glutathione peroxidase decrease in serum of tau transgenic mice expressing P301L mutation [179,180,181].

#### 4.1.6. Extra Virgin Olive Oil (EVOO)

Thanks to its bioactive components (over 200), EVOO is considered both a nutraceutical and functional food. It characterizes the Mediterranean countries and, since the time of Ancient Greeks, EVOO has been considered an “elixir of youth and health” due to its nutritional properties [182]. Actually, many studies support the idea that EVOO represents the best dietary choice for pursuing a healthy ageing. Its positive effects mostly depend on polyphenol content.

The main constituents of olive oil are triglycerides (98–99%), in particular the three main fatty acids, i.e., oleic acid, palmitic acid and linoleic acid [183]. The remaining unsaponifiable fraction contains about 230 components, including lipophilic phenols (tocopherols), whose levels fall as olives mature, sterols, colored pigments, chlorophylls and carotenoids (the most important is β-carotene), alcohols, waxes, aldehydes, esters, ketones and phenolic compounds (hydrophilic phenols). Obviously, the nutritional and healthy effects of olive oil have been ascribed to the unsaponifiable fraction (about 2% of total) [184]. Polyphenol content is greatly influenced by the variety and the geographical origin of the olives and it also depends on several variables linked to agronomic cultivation practices. Thus, not all the olive oils have the same nutritional value. Notably, a high phenolic amount is accompanied by an increase in the nutritional quality of olive oil. It follows that lower quality olive oils lose antioxidant activity, vice versa an olive oil rich in phenolic compounds is able to increase its antioxidant and anti-inflammatory effects, according to [185].

Oleuropein, hydroxytyrosol and oleocanthal are the most noteworthy phenolic compounds found in olive oil. They act as antioxidant compounds counteracting ROS formation and exerting a protective action against the amyloid plaque generation and deposition [185,186].

In addition to the in vitro antioxidant property of olive polyphenols, current evidence supports the concept that these compounds may also confer an indirect antioxidant protection in vivo by modulating the expression of genes involved in the intracellular redox status control [185,187].

However, although several studies have shown the effects of olive and EVOO polyphenols on cell lines or animal models, few studies have been conducted in humans both in vivo and ex vivo. Further, it is clear that in vivo antioxidant effects of EVOO are strictly linked to its bioavailability from the diet.

Observational studies, indirectly confirming the neuroprotective effects of olive oil, showed that MD is linked to a reduced risk of AD [178,188], independently from other factors such as sex, education or APOE gene allele. This diet has been promoted by researchers, clinicians and institutions as a model for healthy eating mainly because it provides a significant source of antioxidants [189]. It is the traditional dietary pattern followed by people residing on the shores of the Mediterranean Sea, where populations are characterized by simple food habits as high intake of whole cereals, fruits and vegetables, legumes and fish, olive oil as the common source of fats and moderate, regular wine drinking. MD is low in saturated fatty acids and trans fatty acids due to its low content of animal meats and processed foods.

### 4.2. Nutraceutical Compounds against Aβ and Tau Toxicity and Aggregation

Loss of brain function and reduction of brain nerve tissue in AD are closely related to amyloid plaques and NFTs (see Section 3.1).

Given to the complex mechanism of dementia, current therapeutic attempts mainly focus on inhibiting amyloid production and reducing the negative effects of amyloid and neurofibrillary tangles accumulation [190]. There is strong scientific evidence that eating blueberries, blackberries, strawberries and other berry fruits has beneficial effects on the brain and can help preventing memory loss and other age-related changes [13,191,192].

Based on the oxidation state of the central ring, the flavonoids are further divided into 6 groups: anthocyanins, flavonols, flavanols, flavones, flavanones and isoflavonoids. Science has recognized the ability of these substances to contrast degenerative processes, even those involved in the onset of AD.

Recently, Li et al. [193] have revealed that anthocyanins-responsible for the red, purple and blue color of many fruits and vegetables [132]—increase the activation of the FKBP52 protein, which has an affinity for the phosphorylated tau protein and prevents its aggregation. Another strong candidate to provide neuroprotection in AD is blackcurrant, in which anthocyanin represents about 80% of the total amount of quantified compounds and has been shown not only to inhibit the formation and extension of Aβ fibrils, but also to destabilize the preformed Aβ fibrils in vitro [132], thus neutralizing the toxic effects of amyloid and protecting nerve cells. In addition, also *Ginkgo biloba* extract, the Chinese herb studied for its potential benefit in the treatment of AD, seems to inhibit Aβ aggregation [194] and prevent Aβ-induced neurotoxicity in vitro [65].

#### 4.2.1. Genistein

GEN, exhibiting a strong antioxidant property (see above), is effective against Aβ toxicity, but the underlying mechanisms are unclear.

An in vitro study with hippocampal neuronal cells revealed that GEN can significantly attenuate the formation of Aβ-plaques by both downregulation of BACE1 activity and simultaneous upregulation of the enzyme-secretion. In addition, exposure to GEN (0.375 g/mL) activates the PKC signaling pathway and reduces the formation of presenilin, which is responsible for splitting the APP [195].

According to [147], GEN appears to act as a β-secretase inhibitor and a α-secretase promotor, thus decreasing Aβ synthesis and senile plaque formation.

In the study of Ye et al. [196], conducted on an animal model, rats were fed with GEN for 7 days, prior to the intracerebral injection of Aβ to emulate conditions similar to AD. GEN treatment reduced neuronal damage and lowered phosphorylated tau levels in the hippocampus.

GEN supplementation significantly reduces Aβ accumulation and tau hyperphosphorylation, two main characteristics associated with AD in ApoE -/- mice fed a High-Fat Diet [146].

#### 4.2.2. RSV

RSV (3, 4′,5-trihydroxystilbene) is another compound endowed with the potential to reduce Aβ aggregation and toxicity in cell culture by stimulating proteolytic cleavage of the toxic AD protein [140] (Figure 3).

Dietary RSV is essentially derived from grapes and wine [197], but it is also present in other dietary sources, including peanuts pistachios and blueberries [171].

In particular, Fu et al. [198] showed that along with the disruption of the prevalent Aβ plaques, RSV may inhibit the formation of high molecular weight Aβ oligomers by preventing the accumulation of the low molecular weight peptides, thereby reducing the progression of AD. The latest papers reveal that cerebrospinal fluid (CSF) levels of Aβ40 and Aβ42 were significantly decreased as a result of 52-week RSV supplementation [199].

In a transgenic model of AD, Karuppagounder et al. [200] reported that daily intake of RSV decreased Aβ plaque in the CNS and the mayor changes were observed in medial cortex, striatum and hypothalamus. In SH-SY5Y cells, RSV inhibits the activity of BACE1 enzyme, thereby inhibiting the cleavage of APP [145]. Another study, having observed that RSV effectively reduces the cleavage-mediated activation of APP and promotes peptide clearance [201], confirms that RSV can be efficient in reducing the formation of insoluble Aβ plaques in rat hippocampus [140].

It seems that RSV could act as an anti-Aβ compound via different mechanisms: *(i)* it is a more potent inhibitor of Aβ42 aggregation compared with other plant-derived antioxidants (RSV > catechin > curcumin > piceid > ginkgolides) and *(ii)* it reduces Aβ plaque formation in a transgenic model of AD [171].

Moreover, the activation of SIRT1 seems to be its determinant property [140]. The relationship existing between SIRT1 and AD is crucial as highlighted by a study showing that the SIRT1 serum concentration in AD patients was significantly lower than that found in a cohort of healthy subjects [202]. Consistently, several following studies suggest that activation of RSV-dependent SIRT1 represents a promising approach to prevent amyloid deposition and neurodegeneration in AD [140]. Similarly, RSV also induced an increase of SIRT1 in a mice model [203] and SIRT1 overexpression decreases Aβ production [140]. Such an effect may represent an interesting therapeutic approach to block the neurodegeneration and the cognitive impairments caused by the disease. About this, mice overexpressing SIRT1 exhibited reduced brain inflammation due to its action on tau phosphorylation and reduced cognitive defects that were specific to the APP transgenic mouse [140].

Based on the opposite findings of Wightman et al. [204], which reported that no cognitive performance improvement as a result of RSV supplementation occurred, it may be speculated that the observed effects in AD patients can be associated with reduced Aβ synthesis and/or neurotoxicity rather than with the influence of RSV itself on cognitive or mental performance.

In summary, the altered APP processing and anti-inflammatory and antioxidant effects of grape supplementation (containing RSV, catechins, flavonols and phenolic acids: ferulic, gallic and caffeic acids) in an AD animal model were traced back to its ability to improve cognitive deficits [205].

Interestingly, RSV administration has also been found to be beneficial against tauopathy which contributes to AD pathogenesis. In fact, in a mouse model of AD, RSV reduced both Aβ burden and tau hyperphosphorylation [171] as well as the accumulation of the tau protein by upregulating the endogenous factor BAG-2, a member of the family of the molecular chaperone regulators [206].

However, other clinical trial have not confirmed some of the RSV effects observed in preclinical studies, thus showing that any indisputable relationship between RSV treatment and the level of tau protein or hyperphosphorylated tau 181(ptau181) is still lacking [160].

#### 4.2.3. Curcumin

By virtue of its antioxidant, anti-amyloidogenic and neuroprotective activity, CUR shows prophylactic activity against AD, and then, it is considered one of the most promising drug candidates to test in future studies.

The formation and aggregation of the Aβ protein, implicated in the progressive degeneration of brain cells that characterizes this form of dementia, would be hampered in presence of CUR [130].

In vitro studies have showed attenuation by CUR of Aβ peptides aggregation in senile plaques in vitro and in cell cultures [refs see 167]. Other in vivo studies also showed that CUR not only prevents aggregation of new amyloid deposits, but also promotes disaggregation of the existing ones and even reduction in size of the remaining deposits [207]. Consistently, it has been observed that systemic administration of CUR to transgenic mice for one week not only reduced existing amyloid plaques, but also caused a light but significant reversal of structural changes in dystrophic dendrites, thus suggesting that CUR reverses existing amyloid pathology [208].

Other studies highlight the capability of CUR to bind with metal ions, i.e., Cu (II), Zn (II), etc., abundant in the synapse regions, as the main mechanism underling its neuroprotective property. Therefore, these metal ions could play a critical role in Aβ aggregation as well as in the pathogenesis of AD [145].

Along with attenuating deposition and oligomerization of Aβ, it seems that oral administration of CUR can mitigate the phosphorylation of the tau protein and improve cognition and behavioral impairment in lab animals [145].

Regarding tau protein, it has also been found that CUR binds to NFTs in AD brain and inhibits the accumulation of the protein [75]. An in vitro experiment showed that CUR not only inhibits hyperphosphorylated tau protein aggregation but also disintegrates performed tau protein filaments [209].

The two processes of Tau protein hyperphosphorylation and deposition as neurofibrillary tangles are regulated by several tau protein kinases, the most common of which is glycogen synthase kinase-3 [75]. CUR has been proved to inhibit its activity, thus reducing tau protein dimer formation and hyperphosphorylated oligomerization in aged tau overexpressing transgenic mice [210].

#### 4.2.4. (−)Epigallocatechin-3-Gallate

Interestingly, in a study performed on a mouse model of familial AD, researchers observed that CUR, administered individually, lowered amyloid plaque load, but when administered in combination with (−)epigallocatechin-3-gallate (EGCG), docosahexaenoic acid (DHA) and α-lipoic acid (ALA), fails to show any synergistic advantage [207], highlighting that nutraceuticals containing multiple compounds do not necessarily have additive or synergistic effects.

EGCG, a polyphenol catechin [211], is found in high abundance in tea leaves, including green tea and also as trace amount in apple skin, plum, onion and hazel nut.

Epidemiological studies suggest that drinking tea reduces dementia risk [212]. This is the reason why many studies have therefore focused on the tea catechins searching for pharmacological validation of EGCG in particular.

It has been shown that EGCG crosses the BBB after systemic administration [213], modulates APP cleavage and reduces cerebral amyloidosis in AD mouse models [207], thus preventing neuronal death caused by Aβ neurotoxicity [213]. EGCG pyrogalloyl moiety is considered essential to prevent Aβ aggregation and to reduce plaque load, effects that improve cognitive functions in vivo [171,214].

Between catechins, scavengers of ROS and metal ion chelators, EGCG shows more potent effects due to the trihydroxyl group on the B ring and the gallate moiety at the 30 position in the C ring [212]. These chemical features allowed EGCG to scavenge ROS and to protect hippocampal neurons against Aβ as demonstrated in specific in vitro and in vivo studies focused to analyze cognitive functions and dementia mechanisms [215].

Like tea, also cocoa contains catechins. A 2 year chocolate intake was associated with a lower risk of cognitive decline (41%) in 531 persons aged ≥65 years [197].

However, although antioxidant and other effects of cocoa flavanols are relevant to brain health and cognition in the long term, is the vascular mechanistic action of cocoa flavanols that is taken into consideration to explain their effects on cognitive functions.

#### 4.2.5. Lycopene

LYC, an aliphatic hydrocarbon carotenoid extracted from tomatoes, as well as its metabolites, have been hypothesized to get the same neuroprotective effect of other carotenoids [181].

This hypothesis has been supported by previous studies investigating the role of LYC in neurodegenerative disorders [181]. LYC supplementation improves cognitive performance in Tau transgenic mice expressing P301L mutation, attenuate Aβ-induced cellular toxicity in cultured neurons [179,181], and reduces the activation of caspase-3 induced by intracerebroventricular Aβ1-42 injection in rat hippocampus [181]. Eight weeks of LYC treatment reduces the increase in tau phosphorylation in brain tissues of P301L transgenic mice [179], suggesting that inhibition of tau protein phosphorylation may mediate the anti-AD effect of LYC.

In a mouse model of amyloidogenesis, LYC supplementation was found to reverse lipopolysaccharide (LPS)-induced Aβ accumulation and BACE1 increase, thereby improving cognitive function [216].

#### 4.2.6. EVOO

As previously stated, other compounds having neuroprotective action are the phenolic compounds present in the EVOO, whose usage opens new perspectives in the field of neuroprotection. In addition to its antioxidant activity, Oleuropein, the most noteworthy phenolic compound present in olive oil, has many other beneficial properties, such as protecting nerve cells from neurotoxin-induced apoptosis [217]. It also lowers Aβ levels and prevents its aggregation, as well as reduces the expression of glutaminyl cyclase, an enzyme involved in Aβ synthesis and prevents the formation of toxic tau aggregates [218]. The aglycone form of oleuropein is also able to interfere in vitro with the aggregation processes of Aβ oligomers and Tau protein [185], the two critical steps in the initiation of AD pathology.

In the digestive tract, oleuropein is hydrolyzed into hydroxytyrosol, another phenolic compound, which is also present in free form in EVOO and has a higher bioavailability [217,218]. Hydroxytyrosol has been proved to reduce brain mitochondrial oxidative stress and neuroinflammation in AD-prone transgenic mice by induction of Nrf2-dependent gene expression [185]. Furthermore, St-Laurent-Thibault et al. [219] showed in an in vitro study that hydroxytyrosol protects nerve cells against Aβ-induced toxicity.

A similar antioxidant effect has also been reported for oleocanthal, highlighting its ability to interact with Aβ aggregation states and providing neuroprotective benefits [185]. Rodríguez-Morató et al. [217] showed that oleocanthal may reduce Aβ aggregation and modulate its clearance from the brain. Further, in in vitro experiments, conducted on *Escherichia coli*-derived tau protein, Li et al. [220] showed that oleocanthal prevented the accumulation of the protein in comparison with the control group.

#### 4.2.7. Coconut Oil

In addition to EVOO, coconut oil is also rich in phenolic acids. Recent literature has suggested that the use of coconut oil (extra virgin/virgin), coconut water and coconut cream could have significant positive effects on plasma cholesterol and blood glucose levels, and blood pressure control (BP), all of which are recognized risk factors associated with AD [221].

The main phenolic acids, p-coumaric acid, ferulic acid, caffeic acid and protocatechuic acid, present in coconut oil [222], are known for their antioxidant properties. The hydroxyl group of phenolic compounds may be able to reduce the toxicity of the Aβ peptide [221]. However, despite the encouraging studies mentioned above, the exact mechanisms by which the phenolic group affects Aβ toxicity are currently unclear. Ferulic acid, in particular, is one of the phenolic compounds known to have strong anti-Aβ aggregation properties [223]. Continuative ferulic acid administration reduces the cortical levels of Aβ 1–40 and Aβ 1–42, as well as the IL-1b levels in the APP/PSI AD-model transgenic mice [221] and is able to inhibit deposition of Aβ in the brain [221]. On the other hand, another study found that ferulic acid could not prevent the formation of Aβ fibrils, but only reduce the length of the fibrils [221]. Still, others suggest that ferulic acid may be able to interrupt the elongation process by binding to Aβ fibrils [221].

### 4.3. Nutraceutical Compounds against Neuronal Damage and Apoptosis

As previously discussed (see Section 3.2), ROS-induced oxidative stress increases Aβ production and thus initiates a series of downstream neurotoxic events that result in neuronal damage and then death [140]. Several antioxidant products are used to protect against neuronal dysfunction in AD patients [140], preventing death.

Here, we have attempted to trace back those studies in which the action of bioactive compounds has been analyzed by assessing their effect, specifically on death.

#### 4.3.1. Genistein

As reported above, GEN provides a neuroprotective effect both in vitro and in vivo [151].

In CGCs, where extracellular potassium concentration-dependent apoptosis death is induced, it has been shown that treatments with either GEN or its analogue Daidzein (DZN), used at concentrations commonly found in food, prevent death and result both suitable to achieve neuroprotection. In this study the authors documented that GEN protected neurons by inhibiting the mitochondrial apoptotic pathway (see above) and despite sharing a common antioxidant capacity, only GEN and DZN, but not cathechin (CE) and epicatechin (EC), prevented apoptosis of CGCs, suggesting that flavonoid effect is quite specific. On the other hand, a comparison between GEN and CE showed that GEN could prevent cell death in both CGCs and rat midbrain cultures, while CE was effective only in the latter case [152], suggesting that flavonoid effect may also strictly depend on the experimental system studied.

In addition, other researchers showed that GEN can significantly attenuate the damage of nucleic acids and downregulate the caspase dependent intrinsic and extrinsic apoptotic pathways by activating estrogen receptors (ER), an estrogen hormone (17β-estradiol) responsive intracellular or membrane bound proteins, which are mostly G-protein coupled receptors [145].

Wang et al. [224], using AD rats, observed that after treatment with GEN, hippocampal neuronal apoptosis was significantly reduced and the expression levels of caspase-3, Bax and cytochrome c, key components of the mitochondrial apoptotic pathway [224], were significantly reduced. However, the mechanisms of action underlying GEN neuroprotection are not well understood.

GEN has also been reported to reduce apoptosis of hippocampal and cortical neurons in vitro and to cross the BBB to influence brain morphology and reduce Aβ toxicity in the brain [224].

#### 4.3.2. RSV

Apart from having a strong ability both to remove free radicals and to reduce the accumulation of β-amyloid and tau proteins, as above mentioned, even RSV is able to reduce inflammation and protect neurons from death, as demonstrated by in vivo experiments on animal models [136]. RSV treatment proved to be effective in reducing neuronal apoptosis and improving the spatial memory with concomitant decrease in Aβ peptide accumulation and lipid peroxidation in the hippocampus [140]. In addition, RSV may also modulate important glial functions, including glutamate uptake and GSH activity, as well as improve functional recovery, and decrease both DNA fragmentation and apoptosis [140]. This overall evidence suggests that RSV can be considered an effective compound in a strategy for AD therapy and, recently, Ko et al. [225] demonstrated that RSV supplementation prevents the initiation of apoptotic pathway and neuronal loss by lowering oxidative stress.

As already mentioned, the key neuroprotective mechanism of RSV in AD seems to be closely related to SIRT1 activation [140], although the precise mechanisms linking RSV to SIRT1 overexpression and neuroprotection are still unknown.

RSV can also effectively suppress the apoptotic activities of both p53 and forkhead O (FOXO) via SIRT1 overexpression, thus representing a promising approach to preventing neurodegeneration in AD [140] (Figure 3). Parker et al. [226] showed that one wine glass/day brings enough RSV (500 nM) to combat neuronal dysfunction due to SIRT1 activation in AD. More recent studies point out that a higher red wine intake reduces AD risk in men, while increases risk in women [227]. Although the effects of wine and grapes on dementia risk are mixed, potential benefits have undoubtedly been attributed to RSV [171].

Further information on RSV neuroprotective properties are provided by several studies [63,160].

Recently, the protective mechanism of RSV has also been associated with mitophagia [140]. Whatever the target of the beneficial effect of RSV, it is out of doubt that its effectiveness may reside in its ability to penetrate the BBB which allows RSV to have a high neuroprotective capacity, even when administered at low doses.

However, despite the high neuroprotective potential of RSV demonstrated in several in vitro studies, the major limitation is the lack of any information from clinical studies correlating SIRT1 activation with slowdown in the development and progression of AD. Most human studies establish a link between consumption of foods rich in RSV and reducing the incidence or prevalence of AD, as well as improvement in learning, memory, visual and spatial orientation and social behavior.

In addition, it seems that the neuroprotective effects of RSV are mainly short-lived, varying according to dose, dosage form, duration of treatment, pharmacokinetic and pharmacogenetic parameters, etc. This leads to the conclusion that further clinical trials are needed to substantiate the neuroprotective effects of RSV and its likely mechanisms of action in the human body and before allowing RSV to be implemented in AD fighting strategies. However, it is undisputed that RSV is a precious molecule to be used in health promotion, not only for its antioxidant activities but also for its anti-inflammatory and neuroprotective properties.

#### 4.3.3. Lycopene

The in vivo effects of LYC, a carotenoid commonly present in the human diet, are beneficial for amelioration of neuronal damage. Long-term LYC treatment reduces caspase-3 activation in the hippocampus of rat treated with intracerebroventricular Aβ1-42 injection [228], endorsing a neuroprotective effect of LYC in AD. That LYC is potentially able to prevent neurodegenerative processes has been further proved by Hwang et al. [229]. In an in vitro study performed on human neuronal cultures exposed to Aβ, the authors showed an increased survival rate and decreased apoptosis in cells previously treated with LYC.

#### 4.3.4. EVOO

Several studies stated that also EVOO and table olives phenolic compounds exert neuroprotective activity [185].

### 4.4. Nutraceutical Compounds against Memory Loss and Cognitive Decline

AD is commonly diagnosed in people over the age of 65 [132,177], who develop a progressive pattern of cognitive and functional disorders [132] that gradually increase as the AD progresses.

In an observational study, Devore et al. [230] prospected that an increased intake of blueberries and strawberries could be associated with slower rates of cognitive decline in human subjects over the age of seventy, suggesting the potential protective role of berries on different cognitive functions. Thus, it seems that flavonoids present in different varieties of berries may be useful to delay neurodegeneration, to improve memory and cognitive function, and finally to prevent pathological degenerative processes in the brain [171]. Blueberries have been the subject of several studies and one of them revealed that a 2% blueberry diet in older rats F344, performed for 8-10 weeks, improved spatial memory, a result related to the concentration of anthocyanins found in the cerebellum, striatum, cortex and hippocampus [231]. It is also indicative that oral bioavailability is adequate to mediate cognitive effects. A separate study concluded that a blueberry extract (*Vaccinium ashei*; synonymous with *Vaccinium corymbosum*) orally administered to mice for 30 days improved long-term memory [171]. Other fruits having anthocyanins, evaluated for the effects against cognitive decline, include cherries (*Prunus* species; Rosaceae); in short-term randomized trials, the intake of cherry juice improved cognitive functions in patients with dementia [171].

We take this chance to emphasize that, about bioavailability, in vitro BBB models show that some flavonoids and their metabolites can be transported across hCMEC/D3 and RBE-4 cells [171]. Rat RBE4 and human hCMEC/D3 cells [232] are the best characterized among the brain endothelial cell lines widely used in cell culture-based BBB models, useful tools for screening of CNS drug candidates. Usually, cell sources for BBB models include primary brain endothelial cells or immortalized brain endothelial cell lines [232].

The hydrophilic character of some berry flavonoids has raised questions about the relevance of in vitro studies, which bypass the effective bioavailability and metabolism of these compounds in vivo. In fact, dietary phytochemists may not cross the BBB and may not represent the metabolites that can reach the CNS and have direct effect on cognitive functions. One of the reasons for the low number of CNS active drugs in clinical use is just the restricted penetration of most drugs across the BBB [233], the major obstacle to be overcome to reach the CNS.

#### 4.4.1. Genistein

Even isoflavones present in soy are capable of improving cognitive function. First among all, is GEN. As promising compound in the prevention of neurodegeneration, GEN appears to have an advantageous effect in improving learning and memory [145]. The Morris water maze revealed a substantial shortening of escape latency by GEN in AD rats [224]. In addition, behavioral tests, performed on rats that were given GEN for 7 days before they were processed in order to emulate conditions similar to AD, revealed an improvement in their memory and learning ability when matched to rats that had not received GEN [196]

The neuroprotective properties of this isoflavone are supported by the observations of Asian populations, whose traditional diets contain large amount of soy products. Ozawa et al. [234] analyzed nutritional data compiled in 15 years by 1006 old Japanese individuals aged 60 to 79 years and concluded that a diet rich in vegetables, algae, dairy products and soybeans was related to a lower risk of dementia. In particular, the risk has been reduced by two thirds in people who more closely adhere to this nutritional model, further suggesting that diets rich in soy products, containing GEN, could protect the CNS [234].

In addition to this, numerous preclinical studies indicate promising therapeutic activity of GEN against the pathogenesis of AD. A study revealed that GEN at 10 mg/kg dosage can inhibit Aβ-plaque formation in the brain and improve learning and memory deficits through modulation of the oestrogen signaling cascade in rats intoxicated by Aβ.

All together these results suggest that GEN, reducing neuronal loss in the hippocampus, ameliorates learning and memory capacity.

#### 4.4.2. RSV

Most human studies establish a link between consumption of foods rich in RSV and the improvement in learning, memory, visual and spatial orientation and social behavior.

A lower risk of dementia has been observed in subjects who drink moderate amounts of red wine compared to teetotalers [136]. Oral administration of RSV to an experimental mice model of AD showed that it improved the cognitive impairments and memory deficits associated with the accumulation of Aβ [145] (Figure 3). It also proved to ameliorate impaired spatial orientation and memory functions in C57Bl/6 mice by reducing the vascular anomalies in the hippocampus region [145], a critical brain zone for cognitive and memory functions and a very sensitive area in AD [140].

Clinical studies have shown a cognitive improvement in RSV-treated groups [199], contrarily Farzaei et al. [235] reported no evidence that RSV affects memory or cognitive performance in healthy subjects, thus suggesting that there is no consistent data towards cognitive performance improvement as a result of RSV supplementation [204]. At a first glance, this could be discouraging; however, it must be taken into consideration that the effects observed in AD patients were due to a reduced Aβ synthesis and/or neurotoxicity rather than to the direct effect of RSV on cognitive or mental performance.

In fact, it seems that the RSV neuroprotective mechanism which activates SIRT1 [140], contributes to improved cognitive function [140] (Figure 3). In the adult brain, the absence of SIRT1 expression in hippocampal cells is correlated with compromised cognitive abilities, including immediate memory, classical conditioning and spatial learning [140]. Consistently, a significant reduction in hippocampal neurodegeneration, associated with a decrease in SIRT1 acetylation [236,237], was observed after intracerebroventricular injection of RSV in an animal model. In addition, other studies showed that in mice overexpressing SIRT1 the cognitive defects, specific of the APP transgenic mouse [140,238], were reduced.

#### 4.4.3. Curcuma Longa

A polyphenol contributing to the limitation of cognitive deterioration and behavioral symptoms typical of AD is the *Curcuma longa* which plays an active role in therapeutic strategies aimed at preventing or delaying its onset [239]. Observational studies suggest that curry, containing an elevate amount of CUR, promotes the maintenance of mental functions. In particular, in an epidemiological study of 1010 subjects aged 60–93, Ng et al. [240] observed that persons consuming curry “occasionally” and “often to very often” have significantly better cognitive test scores than participants using it “never or rarely”.

Consistently, also Dominguez and Barbagallo [197] observed that people often consuming curry have a lower incidence of AD and consistently healthy older people in these populations have better cognitive functions.

The effect of CUR on mental functions was also analyzed by Small et al. [168] who demonstrate that significant improvement in attention and memory goes along with decreased aggregation of Aβ and tau protein in hippocampus, indicating a strong neuroprotective effect of CUR [168].

A critical observation to the use of *Curcuma longa* in neurodegenerative diseases is linked to its reduced overall bioavailability, a pitfall only partially circumvented by the possibility of using high dosages of active principle, in relation to its very low toxicity. In fact, on one hand the hydrophobicity of CUR favors the passage across BBB and the subsequent accumulation in the brain, on the other hand, CUR shows an extremely low bioavailability, mainly due to its poor solubility in water, its poor stability in solution, initial intestinal passage and rapid hepatic metabolization [241].

Surely, the preclinical evidence of the effects of CUR gives us hope that it will come to support AD therapy, especially as it is a safe and inexpensive substance, easily accessible and able to effectively penetrate the BBB and membranes neuronal.

*Curcuma longa* would thus be framed as a non-pharmacological adjuvant in the treatment of AD, contributing to the limitation of cognitive deterioration and behavioral symptoms typical of this pathology and also playing an active role in strategies aimed at preventing or delaying its onset, especially in those most at risk [239].

#### 4.4.4. Lycopene

Many outcomes indicate that among carotenoids, mainly LYC improves cognition and memory ability of rodents in different pathological conditions. Fourteen days of LYC treatment has been shown to ameliorate learning and memory dysfunction in Aβ1–42-treated rats via inhibiting neuroinflammation [181].

Several neuropsychological tests showed that the participants ingesting additional carotenoids—none only LYC—showed improvements in their cognitive functions and had better episodic memory than the placebo group. Belonging to xanthophylls—accessory pigments of plants belonging to the carotenoid group—astaxanthin has a potent beneficial effect. According to Katagiri et al. [242], it protects nerve cells by lowering caspase-3 activity [178] as well as improving cognitive and learning decay. Like astaxanthin, mental benefits are also provided by lutein and zeaxanthin. Christensen et al. [243] showed a link between higher consumption of these carotenoids and better cognitive functions in AD.

#### 4.4.5. Ginkgo Biloba

Among the plant extracts most used in the therapy of cognitive decline associated with aging and AD [65], those of *Ginkgo biloba*, in use for many years in the clinical field, are certainly worth mentioning. For centuries, *Ginkgo biloba* leaves have been used in traditional Chinese medical practice [65] and is currently used in Europe to relieve the cognitive symptoms associated with a range of neurological conditions.

Its leaves contain a high concentration of terpenes and flavonoids, molecules endowed with several properties: from the capability to modulate brain flow and to act as antioxidant [244], to other beneficial effects found in pre-clinical studies for treatment of cerebrovascular insufficiency, peripheral vascular insufficiency and cognitive impairment associated with aging and AD [244]. Some recent meta-analyses suggest a positive effect (in terms of cognitive performance and behavioral disorders) of the *Ginkgo biloba* extract (EGB-761) in patients with mild cognitive decline or AD [245]; in particular, it seems that therapy has a significant effect only with dosages greater than 200 mg/day [245].

The few data available regarding the intake of flavonols, including quercetin, kaempferol and myricetin, reported beneficial effects on cognitive performance [136]; in particular, Kaempferol has been observed to improve cognitive learning and memory capacity in mice [136]. Moreover, cognitive performance improvement, memory and attention [136] have been constantly observed.

#### 4.4.6. EVOO

It is no wonder that also high intake of EVOO in older individuals with high cardiovascular risk has been found to be associated with better memory function and global cognition [185]. In preclinical studies conducted on aged mice with cognitive and motor disorders, dietary long-term administration of olive oil phenols is able to modify miRNA expression profiles involved in neuronal function and synaptic plasticity [185]. More and more evidence from clinical trials and population studies indicate olive phenolic compounds as key responsible for the MD protective effects against cognitive decline and neurodegenerative diseases such AD, as well as for the improvement of cognitive performance [185]. Consistently, Psaltopoulou et al. [246] indicated that MD correlates with lower odds of developing AD, suggesting that protective effects may be related to the antioxidant and anti-inflammatory effects of the ingredients of MD. They also claimed that this type of diet can be effective in preventing CNS degeneration. Recently, data are emerging showing that the MD could help in delaying the progression of cognitive decline while improves memory and cognitive function [247].

## 5. Nutrients Modulation of Gut Microbiota as Therapeutic Strategy for the Treatment of Alzheimer’s Disease

Although the communities of around trillions of commensals (such as bacteria, archaea, protozoa and viruses) present in our gastrointestinal tract—the gut microbiota—are stable, they can be quickly influenced and/or altered by aging, illness as well as by common human actions-such as antibiotic exposure, lifestyle and dietary changes. In this framework, the loss of microbiota homeostasis, or dysbiosis, is largely accepted to contribute to the clinical progression of several age-dependent neurodegenerative disorders, including AD [123,125,248,249,250,251,252,253,254,255]. An imbalance intestinal microflora has been found in AD patients with dementia compared with healthy age-matched control subjects [126,256,257,258].

Due to the failure of physiological aging occurring during neurodegeneration, an altered cross-talk between gut microbiota and host occurs with the detrimental shifting toward “unhealthy” population which negatively affects the brain signaling. This situation further exacerbates the disease-associated symptomatology by releasing circulating inflammatory mediators, by activating/modulating oxidative stress and metabolic signaling pathway(s) and by unbalancing the production of key neuroactive molecules. Indeed, gut microbiota is engaged in bidirectional interplay with the CNS by endocrine, neurological and biochemical mechanisms along both direct and indirect pathways. Anatomically, the multimodal and reciprocal gut-brain physiological interaction takes place by means of a complex network including the sympathetic and parasympathetic autonomic nervous system, the hypothalamic–pituitary–adrenal axis system, the immune system and the enteric nervous system. The inflammatory immune mediators such as cytokines and chemokines, the endocrine descending hypothalamic–pituitary–adrenal axis with cortisol secretion, the ascending vagus nerve sensory pathway which connects the intestines to brain, the tryptophan metabolism, the secretion of circulating bacterial metabolites (short-chain fatty acids and neurotransmitters) are all parts of the two-way communication characterizing the microbiota–gut–brain (MGB) axis [259,260,261,262,263].

To all this is added the fact that gut microbiome also participates to epigenetic changes, as interestingly reported Krautkramer et al. [264]. They clarified for the first time the connection between diet and epigenetic mechanisms having as an intermediate step the gut microbiome and the metabolites that this produces depending on the nutrients it finds in the digestive system. Briefly, the authors compared the intestinal microbiome of mice fed on a balanced diet with that of mice fed on an unbalanced diet, i.e., low in fiber and complex carbohydrates and rich in simple fats and sugars. In the two cases the microbiome was found different, in fact with the unbalanced diet, mice produced a lower level of some metabolites than those fed the healthier diet: specifically short chain fatty acids produced by fermentation of fibers by microorganisms. The hypothesis that these metabolites were responsible for the epigenetic changes was supported by the administration of water containing short-chain fatty acids to mice without microbiome. The epigenetic “signatures” were identical to those found in mice fed a balanced diet, demonstrating that metabolites alone can produce the observed epigenetic changes.

Based on these considerations, nutritional approaches that use diet to modulate the composition and diversity of the intestinal bacterial flora are currently under investigation. They represent a novel, attractive AD therapeutic option to contrast the different neurodegenerative aspects associated with this heterogeneous age-related disorder, including neuroinflammation, cerebral amyloidosis and clearance, tau deposition, neurotransmission imbalance, oxidative stress, autophagy-mediated protein degradation, vascular degeneration and neuronal loss [252,265]. To this regard, preclinical and clinical evidences from AD animal models and affected people actually support the use of functional food and/or nutrition supplements as a sort of therapeutical tool to modulate the gut microbiota population. The aim of this approach is to induce the selective growth of specific resident bacterial populations along the luminal/mucosal intestinal tract to alleviate the clinical and histopathological dementia-related signs [266]. Biotherapy based on consumption of natural compounds is actually proved to ameliorate the gastrointestinal comorbidities and, then, the mental health and lifestyle of elderly patients suffering AD opening promising therapeutic and prophylactic avenues for the treatment of this chronic age-related disorder in human beings. In this regard, the nutraceutics from MD have been shown to promote the specific gut microbial profile [267], to reduce the cerebral Aβ accumulation [268] and, then, to exert a beneficial effects on cognitive decline of AD [269].

Here is reported a critical overview of how several bioactive nutrients (pre- and probiotics, omega-3, fatty acids and polyphenols) can influence the gut microbiota and, then, the molecular signaling of the CNS paving the way for a dietary ‘disease-modifying’ strategy for clinical management of AD symptomatology.

### 5.1. Prebiotics and Probiotics

Prebiotic are plant-derived fibers and non-digestible phytocompounds of food which are fermented only in the large colon where they contribute by feeding the proliferation/colonization of “healthy” intestinal bacteria. Probiotic are living microorganisms, generally Gram-positive taxa from the *Lactobacillus* and/or *Bifidobacterium* genus that are naturally created by the process of fermentation in a few foods, like yogurt and kefir.

Regarding the beneficial effect of pre-/probiotics or their symbiotic combination on the AD phenotype, Bonfili and colleagues [270] demonstrated that in 8-week-old male 3xTg mice—a well-established transgenic AD animal model—4-month administration of lactic acid bacteria and *Bifidobacteria* was able to interfere with inflammatory cytokine and gut hormones. In particular, a decrease in pro-inflammatory cytokines (IL1α, IL1β, IL2, IL12, IFNγ and TNFα), a reciprocal increase in anti-inflammatory cytokines (IL4, IL6, G-CSF, GM-CSF) and an increase in neuroprotective ghrelin, leptin, GLP1 and GIP was observed. In turn, the newly acquired equilibrium of inflammatory cytokines and gut hormones positively impacts on the animal behavioral performances (improving memory functions), the amyloid plaque deposition (reducing Aβ load), the impairment of proteasome and the autophagic pathways. Although the authors clearly demonstrated that changes in microbiota can stimulate the cerebral amyloid clearance, whether and how the nutritional regimen affected the tau neuropathology-another AD-associated hallmark detectable in this mouse strain-were not further investigated. More recently, in the same AD animal model, Bonfili et al. [271] more extended their observations and demonstrated that probiotic mixture markedly reduced the oxidative stress by activating brain SIRT1, the NAD^+^-dependent protein deacetylase endowed with neuroprotective action. The dietary-induced stimulation of SIRT1 signaling was able to decrease the ROS levels by increasing the enzymatic activities of catalase, superoxide dismutase, glutathione S-transferase and glutathione peroxidase and promoting the cell survival via the expression of transcription factors p53 and the FOXO family genes. These findings fit well with the positive role of SIRT1 in the autophagy regulation in 3xTgAD mice following probiotic-enriched diet [270]. However, a deep investigation on the actions, at both gene and protein levels, of this deacetylase on other autophagic mediators, such as LC2 and Beclin, is still missing.

Kobayashi and colleagues [272] reported that short-term oral administration of *Bifidobacterium breve* strain A1 in an experimentally-induced AD model, such as mice intracerebroventricularly injected with a mixture of monomeric and/or oligomeric Aβ25–35 or Aβ1–42, ameliorated the mice cognitive function. Although the authors did not investigate whether probiotic was actually able to lower the Aβ accumulation in vivo, their compelling findings revealed that the intake of *Bifidobacterium breve* strain A1 successfully downregulated the hippocampal expression of key inflammation and immune-reactive neuronal genes which are aberrantly activated by Aβ exposure.

Another study performed by Nimgampalle and Kuna, [273] evaluated the consequences of 60 days consumption of *Lactobacillus plantarum* MTCC1325 in rats which were induced to acquire an AD-like phenotype. Results showed that supplementation significantly improved the spatial learning skills, modulated the cholinergic neurotransmission in hippocampus/cerebral cortex regions and attenuated the neuronal degeneration in probiotic-fed rodents.

Similar results were obtained by Chen and colleagues [274] who investigated the effects of short-term administration of fructooligosaccharides (a type of prebiotics stimulating the growth of Bifidobacteria and Lactobacilli) from *Morinda officinalis* (OMO)—also known as Indian mulberry—in rats with AD-like symptoms. After 8 weeks of treatment, the rat memory was significantly improved in correlation with reduction of the extent of neuronal apoptosis, down-regulation in the expression level of tau and Aβ1-42, mitigation of neuroinflammatory response and oxidative stress, modulation in the synthesis and secretion of neurotransmitters (norepinephrine, dopamine, 5-hydroxytryptamine). Moreover, OMO proved to have a positive effect on the energetic metabolism, in particular on Na^+^/K^+^-ATPase levels in brain tissues from treated animals, another parameter known to decline during physiological ageing and AD.

Recently, it has been reported that 8 weeks administration of probiotics (*Lactobacillus acidophilus*, *L. fermentum*, *Bifidobacterium lactis* and *B. longum*) in AD rat model, induced by intrahippocampal injection of Aβ1-42, exerted an improvement in hippocampal-dependent memory and learning abilities with progressive normalization of the oxidative stress biomarkers [275].

In another research, transgenic *Drosophila melanogaster*, characterized by overexpression of human BACE-1 and APP695 genes, was used as a model of AD [276] to analyze the effect of three probiotic strains. Interestingly, after short-term treatment with *Lactobacillus plantarum* NCIMB 8826 (Lp8826), *Lactobacillus fermentum* NCIMB 5221 (Lf5221) and *Bifidobacteria longum spp. infantis* NCIMB 702,255 (Bi702255), increased survival and motility, reduced Aβ accumulation, ameliorated acetylcholinesterase activity and increased mitochondrial electron transport chain activity were detected in connection with the stimulation of broad-acting transcriptional regulatory factor PPARγ.

Abraham et al. [277] investigated, in APP/PS1 transgenic AD mice, the beneficial action of a mixture containing lysates of the probiotics *Bifidobacterium longum* and *Lactobacillus acidophilus,* vitamins and omega 3 fatty acids. After 20 weeks, in comparison to their age-matched wild types, APP/PS1 animals showed an improved cognitive performance in correlation with reduction of Aβ-positive plaques in the hippocampus and increased colonization of administered bacteria in their microbiota, thus indicating that the beneficial effects could be partly due to dietary-mediated alteration of the intestinal microflora.

Recently, the causal correlation between gut flora composition and cerebral amyloidosis [278] has drawn the attention of basic and translational neuroscience research [266,279]. In contrast to preclinical animal model, a large part of the studies involving human beings suffers limitations in experimental design, including the enrolment of young and/or too small cohorts of subjects with poorly-diagnosed clinical dementia, the insufficient profiling of gut microbiota colonization and functionality on stool samples, before and after the treatment, the exclusion of placebo-treated control group and the lack of cognitive outcomes measures [280,281,282,283,284]. Interestingly, following a randomized, double-blind and controlled clinical trial enrolling about 60 AD patients treated with milk either alone (control group) or with a mixture of probiotics (probiotic group), Akbari and colleagues [285] showed that 12-week intake of the probiotic mixture, containing *Lactobacillus acidophilus*, *Lactobacillus casei*, *Bifidobacterium bifidum* and *Lactobacillus fermentum*, significantly improved their memory/cognitive functions and the metabolic status. However, when a similar study was performed enrolling a cohort of severe AD patients at late stage of clinical dementia, neither cognitive function nor biochemical factors were found ameliorated after administration of another multibiotic formulation of probiotic strains, including *L. fermentum*, *L. plantarum*, *B. lactis* or *L. acidophilus*, *B. bifidum*, *B. longum* [286].

Tamtaji and colleagues [287] carried out a randomized, double-blind, controlled clinical trial enrolling 79 patients with AD which underwent dietary co-supplementation with probiotics containing *Lactobacillus acidophilus*, *Bifidobacterium bifidum*, *Bifidobacterium longum* and selenium for 12 weeks. In comparison with AD experimental group receiving only selenium or placebo, the probiotic and selenium co-supplementation resulted in a significant improvement in cognitive function and some metabolic profiles (such as the biomarkers of inflammation and oxidative stress, insulin level, serum triglycerides, VLDL, LDL and total-/HDL-cholesterol).

Of note, the employment of pre- and/or probiotics as nutritional strategy in AD animal models and affected subjects in order to bring symptomatic benefits has been also excellently reviewed by others [252,288,289].

### 5.2. Polyunsaturated Fatty Acids (PUFAs)

The polyunsaturated fatty acids (PUFAs) are key constituents of neuronal cell membranes, crucially involved in maintaining the proper membrane fluidity required for synaptic signaling process to occur. In addition, the dietary PUFAs also influence both the composition and diversity of the intestinal microbiota, thus contributing indirectly to neuronal signaling via modulation of gut–brain communications [251,252,290,291]. Among the nutrients that are considered potential candidates for interventions to delay the AD progression, there are the n-3 long-chain PUFA which include omega-3 (ω-3 PUFA), docosahexaenoic acid (DHA) and eicosapentaenoicacid (EPA) from fish oil.

To date, the intake of ω-3 PUFA, DHA and EPA has been reported to lessen brain amyloid [292], tau hyperphosphorylation and/or somatodendritic redistribution [293,294] and the neuronal loss as well [295] in three different AD animal models. Furthermore, a double transgenic mouse model for AD characterized by the concomitant expression of APP and n-3 PUFAs exhibited an improvement in cognitive and behavioral deficits [296]. Likewise, in 3xTg-AD mice, DHA consumption boosted the cognitive performance and normalized the electrophysiological alterations as compared to age-matched wild-type group. These findings support the neuroprotective action of bioactive naturally occurring ω-3 PUFA in preventing and/or ameliorating the AD-associated cognitive decline in preclinical experimental models but whether the nutritional treatment is actually able to influence the animals’ brain performance by directly harnessing their gut microbiota remains still to be demonstrated.

Conversely in humans, although PUFAs supplementation has been clearly proved to normalize the gut microbiota dysbiosis [297,298,299,300,301], most of the randomized trials have showed no statistically significant delay/mitigation of cognitive decline of AD patients at middle-late stages [302,303,304,305,306,307]. The small sample size which drastically limits the statistical power, the heterogeneity in composition, the dosage and duration of supplementation, the uncontrolled experimental design are only a few of the limitations of these clinical studies whose cognitive outcome on AD development turn out to be insufficient or low-strength. However, when PUFAs are administered to patients suffering Mild Cognitive Impairment (MCI)-prodromal state preceding the clinical AD manifestation of full-blown dementia-dietary treatment actually appeared to be effective in slowing the disease progression likely by increasing the Aβ clearance via stimulation of M2 macrophagy phenotype endowed with anti-inflammatory and optimal phagocytic functions [308,309].

Notably, experimental and clinical studies of PUFAs supplementation on cognition and neuropathology of AD animal models and affected patients has been also in depth reviewed by others [310,311].

### 5.3. Polyphenols

If the interaction between nutrients (micro and macro) and microbiota is very tight, such a relation essentially concerns polyphenols [312]. Polyphenols are a large family of natural compounds produced by plants endowed with scavenging protective properties against the oxidative stress and/or inflammatory signaling occurring in the brain during neurodegeneration. As known (see above, Section 2.1), the bioavailability of this group of compounds is poor—due to low absorption and fast metabolism leading to their rapid elimination from the organism—however, gut microbiota metabolizes them into active and bioavailable metabolites. This is an interesting aspect of the issue; in fact, it is suggested that the composition of bacteria is a key element in the response to polyphenol treatment and that gut microbiota are probably responsible for polyphenols health effects by regulating their bioavailability [312].

As for PUFAs, dietary polyphenol intake modifies the gut microbiota composition—by promoting the selective growth of beneficial bacteria and/or by inhibiting the pathogenic ones—suggesting that these bioactive compounds might also indirectly influence the brain functions [313,314,315,316,317].

For instance, Yuan and colleagues [318] reported that 20 h treatment of several bioactive constituents from pomegranate fruit (*Punica granatum* L.) such as ellagitannins and/or their physiologically relevant gut microbiota-derived metabolites named urolithins (6H-dibenzo[b,d]pyran-6-one derivatives)—a specific subclass of polyphenols—exerted a strong protective effect on neurotoxicity and muscular paralysis in the *Caenorhabditis elegans* AD model, a worm engineered to express a temperature-sensitive human transgene of Aβ. However, it is not clear whether the preventive disaggregating abilities of the urolithins on the neurotoxic Aβ fibrillation is more likely to take part in the overall neuroprotective effect of pomegranate extracts against the AD-associated changes.

Both in animal AD models and in patients, polyphenols, such as CUR, RSV, EGCG, have recently appeared to be promising candidates for nutritional intervention in AD by improving the deficits in spatial cognition with reduction of cerebral amyloid deposition, tau hyperphosphorylation and neuroinflammation (see detailed description in Section 4) [197,199,261,319,320,321,322]. On the other hand, polyphenols are also known to have a healthy, positive effect on gut microbiota composition. This is the case, as an example, of isoflavones, flavanol from cocoa and red wine that proved to increase the presence of the same “healthy” bacteria (Firmicutes and Bacteroidetes) that were found to decrease in condition of cerebral inflammation or amyloid upload [279].

The composition of gut microbiota is very often the cause of some confusing results obtained from clinical trials, because of individual variability. It is suggested that the composition of bacteria is a key element in the response to polyphenol treatment and that gut microbiota are probably responsible for polyphenols health effects by regulating their bioavailability. Unfortunately, the hypothesis that the therapeutical effect of polyphenols on AD patients could be due to a primary action of polyphenols on intestinal microflora is still waiting experimental and clinical evidence.

## 6. Bioactive Compound Actions on Epigenetic Mechanisms in Alzheimer’s Disease

Young and fascinating science, epigenetics—term introduced in the late 1930s by the English biologist, geneticist and paleontologist Conrad Hal Waddington—investigates that part of genetics that affects gene expression, otherwise called “phenotype”. In simple terms, the gene is expressed in one way or another, in health or disease, in relation to multiple factors, including all the interactions that our body has with the internal and external environment.

Epigenetic changes do not involve changes in the DNA sequence but, rather, they involve multiple processes such as DNA methylation, histone code modifications and noncoding RNAs biogenesis to regulate gene expression [323]. These epigenetic factors can determine the selective “ignition” or “shutdown” of genes, also providing an explanation of how the genetic material can adapt, in a short time, to environmental changes.

Therefore, the real matters are not so much our genes, but rather the expression of our genes as a function of the multiple stimulations, modifications and alterations induced by the environment. Epigenetics comes to assume a sort of “positive” and “liberating” value compared to the determinism and “condemnation” of genetics. Basically, it’s not possible to intervene on genes (or at least not completely and not yet), but on gene expression, yes. Therefore, two individuals with an identical genetic makeup can develop different phenotypes thanks to the differences in their “epigenome”; in this sense, the studies conducted on monozygotic twins, demonstrating the presence of an epigenetic drift between subjects with the same genotype, are exemplary.

Ten years ago, the magazine Time dedicated the cover and a service about this theme, by using a significant title: Why Your DNA Isn’t Your Destiny [324]. The message is extremely modern: DNA is not our destiny, so you can do something to change your destiny.

Over the years epigenetics has conquered the limelight in multiple stages, including those of cheap health gurus. In parallel, more and more scientific papers are emerging on the relationship between epigenetics and diseases.

At the beginning of the 2000s, only 3 pathologies were considered as unquestionably linked to epigenetics: Rett syndrome, fragile X syndrome and ICF syndrome [325,326,327]. In the last 10 years, experimental evidence has shown that most of the multifactorial diseases are or could be induced by epigenome alterations, among these, first and foremost, apart from tumors are the neurodegenerative syndromes.

An interesting aspect of epigenetic modifications is that, unlike genetic mutations, they do not involve the nucleotide sequence of DNA and are, by their nature, reversible. In principle, drugs are able to reverse epigenetic defects but not genetic alterations [328]. One of the richest sources of elements, capable of regulating the expression of our genes, is precisely food. The molecular information contained in it, through interaction with the genome, is able to regulate the metabolism, to make us lose weight or gain weight, to determine whether we get sick or remain healthy. Therefore, the different nutrient molecules introduced with food are able to condition the functions of our genes so, in short, food can interfere with the expression of genes.

Notwithstanding it is not clear whether the epigenetic changes observed in AD patients represent a cause or a consequence of the disease, emerging evidence suggests that epigenetic regulatory mechanisms may be important targets in the treatment of AD [323]. Moreover, given the dynamic nature of the epigenetic marks, intense research is carried out to investigate the therapeutic efficacy of compounds exerting epigenetic properties.

Significant and promising is the study, published last year in Brain [329], in which a group of researchers managed to recovery memory in an animal model. By analyzing the prefrontal cortex of mouse and deceased human donors, the authors have identified an epigenetic anomaly able to modify the expression of the genes linked to glutamate receptors. Both in animal model and human AD brains; in fact, these receptors were greatly diminished, thus leading to synaptic loss of function and decreased memory. Once the malfunction was identified, the researchers tried to intervene to restore the lost skills acting on the proteins responsible for the abnormal epigenetic changes which have become the targets of the new treatment. In particular, treatment of FAD mice with specific EHMT1/2 inhibitors reversed histone hyper-methylation and led to the recovery of both glutamate receptor expression and excitatory synaptic function in prefrontal cortex and hippocampus, thus rescuing both working and spatial memory. These results suggested that the altered glutamate receptor transcription, due to an aberrant epigenetic regulation, underlies the synaptic and cognitive deficits in Alzheimer’s disease and opens new perspectives since targeting the histone methylation enzymes might represent a novel therapeutic strategy for this neurodegenerative disorder.

Even more recently, Monti et al. [330] discovered the existence of an association between alteration of the Presenilin1 (PSEN1) gene and AD.

For the first time in human sample, it has been observed that this alteration depends on “non-CpG” methylation, an epigenetic modification capable of leaving a specific “imprint” on PSEN1 and activating a series of molecular mechanisms that induce overexpression of the gene. The data suggest “non-CpG” methylation as a possible biomarker to be monitored both to identify environmental factors capable of triggering the neurodegeneration process and to evaluate the response to a therapeutic treatment for the disease. Consistently, experimental studies, first conducted on a mouse model of AD and then verified in human brain tissue samples, confirmed a significant inverse relationship between PSEN1 gene expression and DNA methylation in AD patients. Interestingly, a blood sample analysis of AD patients, in comparison to healthy subjects, revealed that low levels of DNA methylation are correlated to PSEN1 expression and give a glimpse about a new way of early diagnosing AD by a slightly invasive method.

Recent research is increasingly shedding light in the biochemical, cellular and epigenetic modifications produced by a series of plant polyphenols, with particular emphasis on the olive ones [331]. As widely discussed in the previous paragraphs (see above, Section 4 and Section 5), the antioxidant power associated with these bioactive compounds involves the modulation of the oxidative pathways [331], the direct action on enzymes, proteins, receptors and different types of signaling pathways [331], as well as the interference with epigenetic modifications of chromatin [332] (see below), thus corroborating the hypothesis that epigenetic nutritional research has substantial potential for AD and may represent a window of opportunity to complement other interventions. It has to be considered that nutrient-dependent modification of the epigenome is an exciting field of research because diet is the “environmental” factor to which the entire population is daily exposed also according to the lifestyle choices.

Nutrients can modify epigenetic marks; however, the impact of diet on chromatin may not always be direct. There are some suggestions that the influence can be mediated by the microbiome, whose composition depends on age as well as on the status of health or disease [30]. As discussed in the previous section (Section 5), nutrients are involved in shaping the microbiota composition, responsible of health span and longevity [30]. The putative links between diet, microbiome and epigenetic mechanisms involved in AD—according to the available literature data—will be discussed in this section. Since DNA methylation is the most studied epigenetic modification, particular emphasis will be given to those food or nutrients endowed with the potential to modulate DNA methylation rather than other epigenetic traits. In addition, although the link between diet and epigenetic mechanisms is more apparent for dietary methyl donors (e.g., folate, choline and betaine) [333,334], it is now widely appreciated that also other dietary molecules can modify the epigenome [14].

### 6.1. Impact of Dietary Factors on DNA Methylation

One of the most common epigenetic mechanisms is DNA methylation which occurs when a methyl group—donated by the S-adenosylmethionine (SAM)—is added to the 5-position of the pyrimidine ring of cytosine, leading to the production of 5-methylcytosine (5mC). The methylation on cytosine residues depends on DNA methyl transferase (DNMT) activity [335]. It occurs predominantly at cytosines preceding a guanine nucleotide (CpG sites) and usually leads to the inhibition or repression of gene transcription. In general, DNA hypomethylation is associated with an increase in gene expression (“ignition”), while hypermethylation leads to gene silencing (“shutdown”).

DNA methylation changes are present in AD-related genes; some of these genes are hypermethylated (MTHFR, Neprilysin, MAPT, APOE, SORB3), while others—including those required for the production of the Aβ peptide—have been found to be hypomethylated (APP, BACE, PSEN1, PP2A, S100A2, CREB5) [336,337]. Methylation changes in post-mortem AD brain samples also affected genes involved in neurofibrillary tangle formation (MAPT and GSK3β) or associated to the late-onset AD forms (APOE) [336,338,339].

It was found that DNA methylation in the peripheral blood mononuclear cells of late-onset AD (LOAD) patients was higher as compared to healthy subjects, and these higher levels were associated to the APOE ε4 and APOE ε3 alleles, indicating that global DNA methylation in peripheral samples is a useful marker for screening individuals at risk of developing AD [340,341]. A hypermethylated CpG island is present within the APOE gene. The APOE4 sequence may influence the epigenetic methylation of the 31-CpG island, since the APOE4 allele can undergo a C to T transition that determines a loss of a methylatable CpG unit [337]. Not only individuals carrying APOE4 allele show a dose-dependent risk but the relative mRNA level of APOE4 is increased in AD patients compared to controls, indicating that the variability in the neuronal expression of APOE contributes to increase the disease risk [337].

Several other genes, involved in LOAD susceptibility, neuronal function or other AD-related pathways, have been proposed as potential epigenetic biomarkers of the disease in either blood or neuronal DNA samples [140], but unfortunately, most of these studies were limited in sample-size, and the results were often conflicting or lack replication, so that the clinical utility of those biomarkers is to date uncertain [342,343].

Besides studying the 5mC levels in AD, researchers have also begun to analyze the role of 5-hydroxymethylcytosine (5hmC) in health and disease. 5hmC is another epigenetic hotspot which is produced during the demethylation of 5mC. In particular, 5hmC is considered to be a key epigenetic mark in brain development and neurological disorders [344,345] leading researchers to postulate that age-associated neuronal 5mC/5mhC alterations may be involved in the early and late phases of AD.

Low levels of 5hmC have been detected in several brain regions of late stages AD patients, such as in the neocortex, hippocampus, entorhinal cortex and cerebellum [346,347,348]. On the contrary, 5mC levels were increased in the hippocampus of aging mice, but reduced in APP/PS1 transgenic mice, as well as in the hippocampus, enthorhinal cortex and cerebellum of patients with AD [349]. In addition, Tohgi et al. [350,351] found that some cytosines, particularly those located in the promoter region of the APP gene, between nucleotides -207 to approximately -182, were mostly methylated and their demethylation caused Aβ deposition in the aged brain. Afterward, studies in neuronal cell cultures and animal models suggested that environmental perturbations, including deficiency of B-group vitamins or lead exposure, may induce methylation changes in genes required for Aβ peptide synthesis [336].

In addition to methylation changes occurring in APP gene, also NEP genes, which encode a protease controlling Aβ degradation, and SORBS3 genes, which encode a cell adhesion molecule, are hypermethylated [352]. In addition, abnormal methylation levels were found in the promoter regions of tau phosphorylation-related genes [347]. For example, analysis of DNA methylation in the promoter region of the GSK-3β gene from the AD post-mortem prefrontal cortex tissue indicated that the GSK-3β promoter region is methylated at low levels during early AD development. Therefore, the mRNA of GSK-3β is upregulated during this period while the GSK-3β protein levels remained unchanged [353]. About that, it was reported that vitamin B deficiency can lead to low levels of cytosine methylation in the GSK-3β promoter region and hence modulate the GSK-3β overexpression [347].

Recently Martínez-Iglesias et al. [354] analyzed the content of 5mC and 5hmC, and the expression of DNMT3a-responsible for de novo methylation [354] showing that global 5mC and 5hmC levels are significantly lower in AD mouse model as compared to healthy animals. Conversely, other studies reported no significant differences between healthy and AD brain samples [354], or even an increase of methylation and hydroxymethylation levels in different regions of the AD brain [354].

The reasons for all the above contradictory results can be many, such as the different brain areas that were analyzed or the heterogeneity of the pathological diagnosis of the analyzed samples. In the case of blood samples, e.g., some studies were performed in serum and others in leukocytes. In addition, the number of samples studied was often too low to obtain conclusive results and the level of global methylation changes with age [354].

Regarding DNMT3, a gene closely related to learning and memory functions [354,355], the authors observed a significant reduction of over 40% in DNMT3a expression in the brain of 3xTg-AD mice. Further evidence about the importance of DNMTs in the aging process is the fact that the hippocampus of aged mice shows a decrease in the expression of DNMT3a while its overexpression can reverse spatial memory deficits [354]. Consistently, a decrease in DNMT expression in neurons and hippocampus of AD patients has also been described [354]. On the other hand, DNMT1 expression, at both mRNA and protein level, is increased in blood mononuclear cells of LOAD patients [340]. Once again, the differences between studies may be explained by the fact that different cells or regions have been analyzed with different methodologies. Indeed, the study of Di Francesco et al. [340] was performed only in patients with late-onset AD, whereas the Martínez-Iglesias et al. [354] study included different types of dementia in different stages of the disease.

Based on the observation that the levels of 5mC and DNA methyltransferase decreased in AD patients, it has been suggested that global DNA hypomethylation is associated with AD [355]. Furthermore, the methylation level of essential substrates, such as the universal methyl donor SAM, the S-adenosylhomocystein (SAH) and folate, were also decreased in human AD brains [356].

In an early study, SAM levels have been found to decrease in post-mortem AD patients [357]. It is worth noting that SAM, maintaining the appropriate methylation of genes involved in APP processing, keep them silenced and thus prevents Aβ formation and accumulation [358,359]. Lower bioavailability of SAM causes changes in the expression of genes involved in APP metabolism, resulting in increased production and/or accumulation of Aβ peptide.

Altered folate/methionine/homocysteine (HCY) and SAM metabolism has been suggested to be related with AD onset [359]. Consistently, Fuso et al. [360] have reported that reducing folate and vitamin B12 in culture medium of neuroblastoma cell lines can cause a reduction in SAM levels which in turn results in increased PSEN1 and BACE levels and Aβ production. The administration of SAM to the culture medium restored normal gene expression and reduced Aβ levels. Vitamin B deficient animal models have shown that SAM inhibits the increase in progression of Alzheimer-like features [361].

More recently, studies indicated that, just like nuclear DNA, the mitochondrial DNA (mtDNA) is regulated by epigenetic factors, via methylation and non-coding RNAs (ncRNAs) in AD [346,362]. This field is referred to as mitoepigenetics. Additionally, it has been established that nucleus and mitochondria are constantly communicating to each other to regulate different cellular pathways. However, little is known about the mechanisms underlying mitoepigenetics. Blanch et al. [363] found a global reduction of 5mC in mtDNA. Analysis of specific loci showed that the levels of 5mC in MT-ND1, a mitochondrial gene encoding a protein of the complex I of the respiratory chain, were lower in the entorhinal cortex of AD patients compared with that in age matched controls. Instead, the level of 5hmC in the mtDNA of AD patients did not change notwithstanding a previous study reported a 5hmC level reduction in aging-associated mtDNA [364]. The discrepancy between Blanch and Dzitoyeva [363,364] studies could be due to the differences between species (humans vs. mice, respectively) and/or the methods used. Dysregulation of mtDNA epigenetic mechanisms could partially explain mitochondrial dysfunction, a recurrent feature of AD [362].

Many bioactive nutrients have been suggested to affect the progression of the disease by interfering with the epigenetic processes which are deregulated in AD [358].

The few data reported above attest a strong association between DNA methylation alteration and AD [365] and suggest that a clear link with nutrition-dependent epigenetic changes exists in AD [365].

In particular, since DNA methylation occurs within HCY metabolism, which uses micronutrients such as folate, methionine, choline and betaine as enzyme cofactors, it is intuitive that dietary methyl-group intake (choline, betaine, methionine and folate) can alter methylation patterns of DNA and histones which in turn results in changes in gene expression.

Then, the role of dietary folate in providing methyl groups required for maintenance and modulation of DNA methylation makes it a nutrient of interest in AD. Folate and B12 vitamins in the form of methylcobalamin contribute to replenishment of cellular SAM [358], the common methyl donor. Folate cannot be synthesized de novo; therefore, it has to be taken by consumption of green leafy vegetables and supplemented foods. Although folate can be generated by microbial activity during digestion, its production is far from the need to meet the metabolic requirements [358].

Fuso and Lucarelli’s laboratory significantly contributed to the research on the epigenetic basis of AD with a series of experiments in a preclinical model of AD mice, coming to formulate a mechanistic process linking nutrition (B vitamin deficiency) to AD-like phenotype through DNA hypomethylation. To do this, TgCRND8 mice, that carry the human mutated APP and produce AD-like senile plaques, were maintained with a diet deficient of vitamins B6, B12 and folate. This treatment induced impairment of the cell methylation potential (MP), deregulation of the DNA methylation and hypomethylation of the gene codifying for the gamma-secretase, thus allowing an increased PSEN1 expression, increased β-amyloid production and greater cognitive impairment. The supplementation with SAM, the methyl group donor, reverted all AD-like features by restoring the MP and the normal PSEN1 methylation [365]. These results highlight a causal connection between the diet, specifically the nutritional content in B vitamins, and the modification of the gene methylation patterns responsible for the disease phenotype.

In addition, many data sustain the remarkable impact of plant polyphenols and other phytochemicals on DNA methylation [358].

A long but surely not exhaustive list of active molecules with epigenetic potential can be identified: LYC, phloretin, hesperidin, CUR, GEN, caffeic acid, isothiocyanates, EGCG and coumaric acid. These molecules, whose source belongs to vegetable world (see above, Section 4), often show different effects on the levels of DNA methylation [365]. However, for some molecules, opposite and apparently contrasting effects have been demonstrated, e.g., CUR seems able to both hyper- or hypomethylating different genes. In other cases contrasting results may be due to the difficulty of standardizing the experimental conditions when working with nutritional supplements; in certain additional cases, the low uptake or bioavailability of a lot of compounds seem to be the problem, as well known for the CUR [366].

However, if specific nutrients showed beneficial effects, others are known for their detrimental effect in contrasting different pathologies through modulation of DNA methylation.

About polyphenols, they are present in fruits and vegetables and are known to exert a protective role in neurodegeneration due to their well-established potent radical scavenging, as well as to interact with epigenetic mechanisms. Tea polyphenols—such as catechin, epicatechin (EC), epicatechin 3-gallate (ECG), epigallocatechin (EGC) and EGCG-bioflavonoids—such as quercetin, fisetin, myricetin, parsley’s apigenin and turmeric’s CUR—have been reported to inhibit DNA methyltransferase (DNMT)-mediated DNA methylation in a concentration-dependent manner leading to demethylation and reactivation of genes previously silenced by methylation [358]. In particular, EGCG, the most effective tea polyphenol, inhibits DNA methylation in both a direct or indirect manner: (i) by forming hydrogen bonds between different residues in the active site of DNMT which results in direct inhibition; (ii) by indirect inhibition of DNA methylation as a result of decreased SAM and increased in both SAH and HCY [367].

Similarly, RSV, a plant polyphenol which is found in peanuts, mulberries, cranberries and mostly in the skin of grapes, inhibits DNMT activity (but in a lesser extent than EGCG) [358] (Figure 3). In addition, quercetin [331] and EVOO polyphenols were shown to activate the Sirt1 path, with possible relevance for AD prevention/treatment [331]. In addition, caffeic acid, a common catechol-containing coffee polyphenol, present also in barley grain, inhibits DNA methylation [368].

Several studies on isoflavones, conducted both in animals and humans, have indicated a positive effect on AD and cognitive function through various mechanisms, such as the downregulation of PSEN1 [348]. A study in rats showed that isoflavones can modulate the methylation of the promoter region of selected genes [369]. In addition, the isoflavone GEN, one of the phytoestrogens found in soybeans (see above, Section 4), is able to regulate gene transcription by affecting epigenetic modifications [358] and its effect on epigenetic alterations has been widely investigated in cancer but not in the field of AD. However, GEN has been proved to reduce DNMT activity by forming a complex with it [331].

Although protective effect of *Ginkgo biloba* extract in AD has been investigated for a long time, its role in the epigenetic alterations correlated to AD pathogenesis has not been examined in detail. Kaempferol is one of the flavonoids found in *Gingko biloba* extract. Kaempferol was shown to inhibit histone deacetylase (HDAC) activity in human-derived hepatoma cell lines and colon cancer cells but its effect on HDACs activity during AD has not yet been investigated [358].

### 6.2. Impact of Dietary Factors on Histone Post-Translational Modification

To better understand one of the molecular mechanisms underlying epigenetics, it must be remembered that DNA is in a “packaged” form due to histones, which are microscopic spheres of proteins having the function not only of packing and organizing DNA, but also of modulating its three-dimensional structure. DNA, so wrapped around a core of proteins, forms the nucleosome with different degrees of compaction, due to modifications of the amino acid residues in the histone tails. By the action of specific enzymes, histones can undergo a series of post-translational modifications (PTMs), including methylation, phosphorylation, acetylation and/or deacetylation and ubiquitination, all processes that can inhibit or increase gene expression. These modifications can alter the accessibility of DNA to transcription regulators by inducing changes to the structural configuration of nucleosomes.

Histone modifications can result from the activity of various enzymes, including histone acetyltransferases (HATs), HDACs, histone methyltransferases (HMTs) and histone demethylases (HDMs) [331].

Alterations in histone acetylation have been reported in studies of AD [352].

The acetylation levels of histone H3 and histone H4 were significantly increased in AD post-mortem brain tissues [370]. Increased histone H3 acetylation is associated with increased levels of β-secretase 1 (BACE1), a protease that cleaves APP in the amyloidogenic pathway [352]. Abnormal histone H4 acetylation is associated with impaired learning and memory functions in AD-associated insults [352]. In addition, non-nuclear histone H1 has been reported to be upregulated in the cells of brain regions susceptible to AD [352].

Histone acetylation is a dynamic reversible process, which is regulated by HATs and histone deacetylases (HDACs), enzymes disturbed in AD-associated process [371].

In AD, class I HDACs, such as HDAC2 and HDAC3, are expressed at much higher levels than the others in the memory associated regions of the brain [372]. Regarding HDAC2, it was reported that AD patients have elevated levels of HDAC2 in the brain [373], and inhibition or knockout of HDAC2 can significantly improve cognitive dysfunction [357,374,375]. Consistently, in AD mice, a significant increase in the expression levels of HDAC2 was found in the hippocampus and prefrontal cortex, while no changes were observed in the amygdala, an area not affected by AD [352]. Moreover, increased levels of HDAC2 and hypoacetylation have been found to negatively correlate with the mRNA expression level of genes associated with learning, memory and synaptic plasticity [341,352,376]. In addition, Graff et al. [377] demonstrated that HDAG2 elevated levels epigenetically block the expression of neuroplasticity genes during neurodegeneration in the CK-p25 AD mouse model; whereas, in another AD mouse model, HDAC2 was found to be strongly expressed in the hippocampus and prefrontal cortex.

Knockdown of HDAC2 abolished the neurodegeneration associated memory impairment. Enrichment of hypoacetylation and HDAC2 negatively correlated with RNA polymerase binding and mRNA expression at the level of those genes involved in the memory and learning processes [377].

Similarly, deletion of HDAC3 in the dorsal hippocampus leads to enhanced long-term memory for object location [341].

In addition to class I HDACs, also class II HDACs are involved in AD. For example, HDAC6 from Class II is elevated in the hippocampus and cortex of AD patients. It co-localizes with tau protein in the AD hippocampus, and tau phosphorylation can be decreased as well as cognitive impairment improved by reducing HDAC6 levels [341,378]. Recent evidence showed that the inhibition of HDAC6 can reverse tau phosphorylation and restore microtubule stability, leading to the normalization of synaptosomal mitochondrial function and synaptic integrity [379,380]. This evidence indicates that HDAC6 inhibitors may be a promising avenue for therapeutic intervention in AD and other neurodegenerative diseases. However, how HDAC6 impacts genes or signaling cascades related to tau phosphorylation is less reported.

The class III HDACs are called sirtuins (SIRT1-7) [339]. The level of SIRT1 is reduced in the AD cortex [341] and the tau acetylation of lysine 28 inhibits tau function promoting its aggregation [341]. In addition, it has been observed that dementia risk increases due to an increase in the SIRT1 level [381]. Regarding this, data associated with epigenetics of AD indicate that low doses of RSV reduce the expression of genes crucial for age-related diseases [348], in particular there is cumulative evidence that RSV activates SIRT1 [348], which leads decreased neuronal loss caused by chronic inflammation.

In conclusion, a range of studies indicates that histone modifications play a vital role in the development of AD. HDACs can both promote and impair memory formation and cognition.

In addition to the involvement of DNA methylation, histone deacetylation could partially explain the etiology and development of AD. The impairment in learning and memory which occurs in the AD mouse model was reversible and improved by treatment with HDAC inhibitors in the study performed by Jeongsil and Young-Joon [357].

As modulators of histone modifications, many dietary polyphenols hold promise to be effective for prevention and therapy of neurodegeneration [382].

For example, EGCG, isolated from green tea, acts as a histone modifier, interfering with the HDAC and HAT activities [331]. It also contributes to histone posttranslational modifications by inhibiting histone methyltransferases [383].

In addition, an increase of acetylation on lysine 4 of histone 5, in the cortex and in the hippocampus of oleuropein (OLE)-fed transgenic mice as compared to untreated littermates, was observed. Recent data show that OLE aglycone given orally for eight weeks to TgCRND8 mice, a model of Aβ deposition, downregulates HDAC2 [384] which is normally upregulated in AD [377]. In these mice, the downregulation of HDAC2 resulted in a significant increase in the level of histone acetylation, in particular of H3 at K9 and of H4 at K5 [384]. Considering that histone acetylation improves cognitive deficits in animal models of AD, this indication is considered a promising novel therapeutic strategy against AD [368].

Oleacein, another EVOO polyphenol, also induces a strong increase in histone and α-acetyl-tubulin acetylation [385].

It has been reported that also garlic and cinnamon polyphenols inhibit HDAC [383].

Recently, it has been defined that polyphenols of cinnamon—renowned from both nutritional and pharmacological points of view for their beneficial health promoting properties mainly attributed to the polyphenolic composition and the volatile essential oils—also inhibited HDAC [386]. Similarly, quercetin, a polyphenolic compound, also found in cinnamon has been shown to increase mRNA expression of SIRT1 in mice [386].

CUR, known to reduce the Aβ production by inhibiting glycogen synthase kinase 3-β-mediated PSEN1 activation, inhibits HDAC isoforms 1, 3 and 8, thus participating in reprogramming neural stem cell–directed neurogenesis [348]. It also inhibits HATs [387]. However, besides CUR, little is known about dietary modulation of HAT activity.

Histone-modifications, as well as miRNA modulation, are also due to flavonols from grape, blueberry, citrus fruits, being mainly involved in NF-kappaB, SIRT1 and MAPKs pathways and inhibit amyloid fibril formation [368].

There is evidence supporting the role of PUFAs, which are fatty acids containing more than one double bond in their molecule and include the subcategory of ω-3 fatty acids, mostly DHA, in slowing cognitive decline [388]. One study on neuroblastoma cells showed that DHA can decrease the levels of HDAC1, HDAC2 and HDAC3. In addition, DHA is involved in histone demethylation processes.

### 6.3. Impact of Dietary Factors on Microrna Regulating Action

An additional genome modulation mechanism is operated by microRNAs (miRNAs). These are small endogenous molecules—20–22 nucleotides in length—of non-coding RNA (i.e., they do not give rise to proteins), single stranded, which modulate the activity of messenger RNA. By binding to the 3′- untranslated regions (3′- UTR) of messenger RNAs (mRNAs), via complementary base pairing, miRNAs can inhibit the translation of target mRNAs, thus modulating protein synthesis.

They regulate more than 60% of the protein-coding genes. Since this complex “network” is finely regulated, even small differences in the expression of miRNAs can lead to the development of various pathologies. Furthermore, since in their mature form miRNAs can be released into the circulation and are extremely stable molecules, their use as biomarkers in the diagnosis and monitoring of various pathologies has been hypothesized.

miRNAs can be used as biomarkers to discriminate different disease forms, staging and progression, as well as prognosis [389]. miRNA dysregulation is implicated also in the pathophysiology of AD [390]. A unique circulating 7-miRNA signature (hsa-let-7d-5p, hsa-let-7g-5p, hsa-miR-15b-5p, hsa-miR-142-3p, hsa-miR-191-5p, hsa-miR-301a-3p and hsa-miR-545-3p) reported by Kumar et al. [389] in plasma could distinguish AD patients from normal controls with >95% accuracy.

Several miRNAs have been identified in vitro to directly regulate the APP mRNA, including miRNA let-7, the miR-20a family (miRs-20a, -17 and -106b), miRs-106a and 520c, miR-101, miR-16 and miRs-147, -153, -323-3p, -644 and -655 [352,391]. Inhibition of miR-101 overexpression reduces APP and Aβ load in the hippocampal neurons. miR-16 targets APP to potentially modulate AD pathogenesis and miR-16 overexpression may lead to reduced APP expression [337]. Specifically, miR-124 is able to alter the splicing of APP exons 7 and 8 in neuronal cells, as well as to regulate the expression of BACE1 [352]. Of note, miR-132 overexpression reduces the protein levels of APP in the brains of senescence-accelerated mouse, while a reduced expression of miR-132 causes APP protein accumulation in AD mice [352].

Several miRNAs also regulate tau metabolism. The miR-132/PTBP2 pathway influences microtubule-associated protein tau (MAPT) exon 10 splicing in the brain, contributing to AD pathogenesis. Both miR-9 and miR-124 are downregulated in AD and might affect tau [392].

Also, miR-15, miR-16, miR-26b, miR-34a, miR-125b and miR-497 have been linked to aberrant tau regulation [352]. Overexpression of miR-16 leads to NFT formation, while a decrease in its expression is correlated to APP protein accumulation [352]. Another epigenetic regulator of the expression of tau, miR-26b, is present at a high level in the brain areas exhibiting AD pathology. In addition, overexpressed miR-125b has been found to induce tau hyperphosphorylation and cognitive deficits in mice, suggesting its involvement in disease pathogenesis [352]. A study by Santa-Maria et al. [393] showed that miRNA-219, which binds directly to the 3′-UTR of the tau mRNA thus repressing tau synthesis, is downregulated in the AD brain. The inhibition of miRNA-132/miRNA-212 promotes tau protein overexpression, hyperphosphorylation and aggregation, resulting in cognitive dysfunction [348]. In addition, in vitro experiments have demonstrated that members of the miRNA-15 family, such as miRNA-15, miRNA-16, miRNA-195, miRNA-497 [348] and miRNA-26a [348], were all involved in regulating several signaling pathways which play a significant role in tau phosphorylation linked to AD development.

In addition to those relating to APP and tau metabolism, miRNAs play an important regulatory function in the expression of other AD-genes, such as β-site APP cleaving enzyme 1 (BACE1), GSK-3β and Sirtuin 1 (SIRT1) [348]. In addition, miRNAs circulating in the peripheral blood and CSF are also considered potential early diagnostic markers and drug targets [348] for AD. Upregulation of miR-34a and miR-181b, and downregulation of miR-9, miR-29a/b, miR-137 and miR-181c, were observed in blood mononuclear cells from AD patients [357].

Little is known about the impact of dietary compounds on miRNA action in AD. However, surely polyphenols hold the scepter as dietary bioactive components also on epigenetic miRNA modulation. A study on apolipoprotein E-deficient mice reported that the treatment with polyphenols affected the expression of five miRNA in vivo [331].

In addition, CUR modulates miRNA expression in ApoE deficient mice [393] and was proved to induce microRNA-22, microRNA-186a and microRNA-199a [348].

More, catechins, presents in green tea, cocoa, blackberries, perform its epigenetic effect not only on DNA methylation and histone-modifications but also on miRNA [368,393].

## 7. The Research Continues

To date, the world of “neuro-nutraceuticals”—as widely observed in this study—offers good research ideas and in the near future it could open interesting windows on effective and safe treatments, even in patients with already diagnosed dementia. Certainly, the preclinical evidence of many food-borne substances-potentially useful in the prevention and treatment of cognitive decline-gives us hope that they will come to support AD therapy. Further studies are needed to confirm their beneficial effect in humans. At the moment, the strongest recommendation in behavioral and nutritional terms—when a good cognitive reserve still remains—is to adopt an active lifestyle together with a diet with the characteristics of the Mediterranean diet (possibly enriched with walnuts and olive oil) [22,23,24,159,188,189,247,269]. Recently, Ravi et al. [394] emphasize how Mediterranean diets, mainly composed of fruits, vegetables and omega-3 fatty acids, stand as valuable, mild and preventive anti-AD agents, able to fight this devastating neurodegenerative disorder through the simple proper modification of our life style.

Certainly, maintaining an adequate diet is important for people of all ages; however, it is particularly important to promote certain nutritional strategies among those of advanced age due to the risk of malnutrition, reduced absorption, loss of appetite and difficulty in chewing, which they can affect correct nutritional status. In fact, in an aging individual, intakes and assimilation of nutrients can be compromised, and may partly explain the conflicting results of clinical studies using nutrient interventions. These strategies are designed to essentially restore gut microbiota diversity in the elderly in order to reduce some adverse consequences of aging. About that, the MGB axis existence and its pathological alterations in subjects suffering MCI and dementia represent the strong rationale to investigate the dietary consumption of bioactive nutrients as novel hopeful therapeutic and prophylactic approach for the AD cure (see Figure 4).

However, up to now, it is not still clear whether alterations in gut microbiota are cause or effect of the phenotypic manifestations underlying the disease symptomatology. Moreover, experimental and clinical results from AD animal models and human beings are not conclusive, mainly due to several experimental limitations, including the lack of well-designed design, the differences in employed methodology (dosage, duration of intervention, combined versus individual administration) and other confounding factors. As the complex interplay between gut microbiota and several dietary nutrients (mainly PUFAs and polyphenols) is not fully clear, additional research is needed to ascertain the potential translation power of its modulation through a personalized diet as an alternative and effective strategy for clinical management of AD.

Therefore, although the role in itself of nutraceuticals in the prevention, restriction and treatment of various diseases is beyond doubt, nevertheless it should be kept in mind that the response of nutraceuticals varies from person to person. The individual’s susceptibility to a particular disease, as well as the assimilation of nutraceuticals, will depend on genetic predisposition, environmental factors and lifestyle.

If we examine the physio-pathological implications of various pathologies (cardiovascular, metabolic, tumor, neurodegenerative, endocrinological, etc.) and the mechanisms that can influence their development and course, we realize how nutraceuticals can influence in a way sometimes determining the fate of the disease itself. This is the therapeutic “new frontier” in the pandemic diseases of the third millennium; but paradoxically, it is a difficult problem to deal with as it involves the most diverse habits of life, different socio-economic statuses and perhaps even economic interests that are not always aimed at encouraging preventive behavior.

What this study reveals is that the word ‘diet’, which derives from the Greek *διαιτα*, does not primarily mean ‘renunciation’, but healthy and proper nutrition, which means considering nutrition not a sack to fill with whatever satisfies your hunger. Is one happier who can afford caviar and champagne every day or one who grasps the value, even physical, of calibrating fresh vegetables with fish, meat, fruit? The first forgets and chases the next pleasure, already codified; the second is still amazed by the pleasantness of the seasonality of foods, which have their own logic, also based on the seasons.

Therefore, life choices are critical to leading a brain-derived or brain-deprived life.

Of course, genetics and many other factors play a role, but modifiable risk factors which are primarily diet and lifestyle are essential if we are to have any kind of impact on diseases such as Alzheimer’s disease.

It is necessary to change direction, and urgently.

## Figures and Tables

**Figure 1 cells-09-02347-f001:**
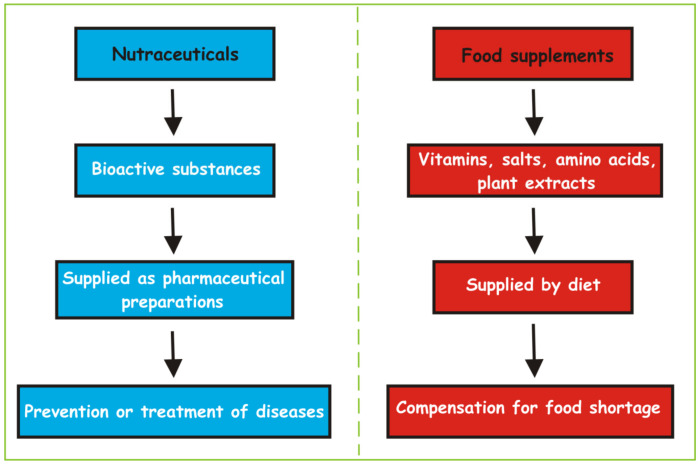
The main and substantial differences between nutraceuticals and food supplements.

**Figure 2 cells-09-02347-f002:**
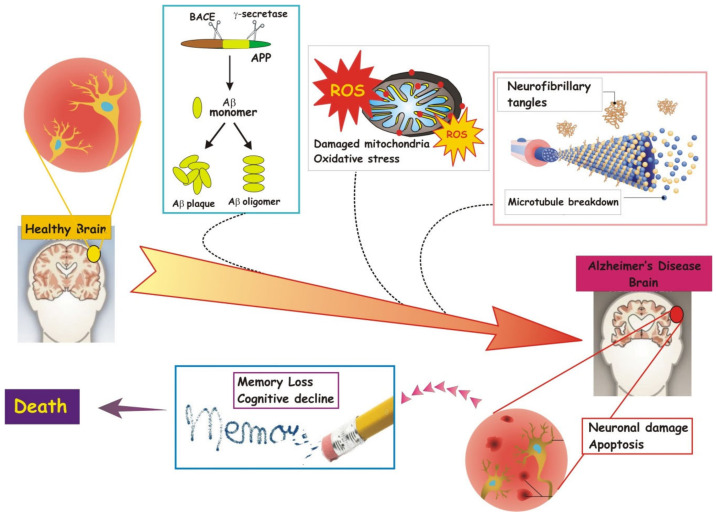
Major events, namely oxidative stress and mitochondrial dysfunction, toxicity and aggregation of Aβ and tau, neuronal damage and apoptosis, memory loss and cognitive decline, characterizing the onset and progression of AD until death.

**Figure 3 cells-09-02347-f003:**
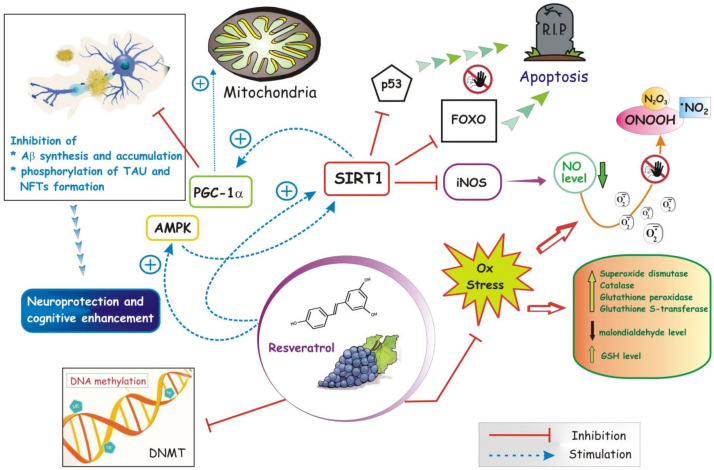
Resveratrol and Alzheimer’s disease. Alzheimer’s disease is a complex neurodegenerative disorder with multiple contributing factors. Some of the pathways affected by Resveratrol are here indicated.

**Figure 4 cells-09-02347-f004:**
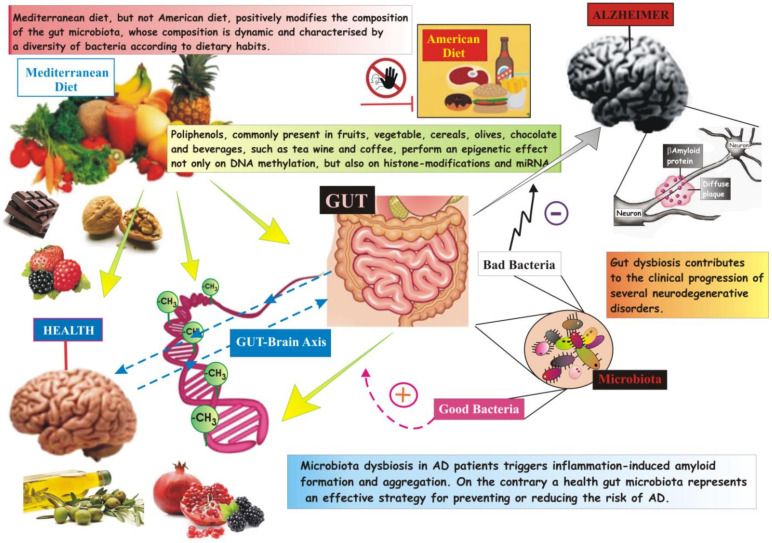
The picture describes that the Mediterranean diet, the richest diet in beneficial and highly nutraceutical substances in the world, may create putative links with microbiota and epigenetic mechanisms by modifying gut microbiota composition; thus, helping us to preserve our mental health, i.e., improve memory, cognitive functions, intelligence and sharpness, avoiding the risk of Alzheimer’s disease.

**Table 1 cells-09-02347-t001:** Nutraceuticals: potential health benefits and disadvantages.

**Advantages of Nutraceuticals**
a) increase the health value of diet b) help to live longer c) help to avoid particular medical conditions d) can be perceived as more “natural” than traditional medicine and less likely to produce unpleasant side effects e) increase the nutritional value of food for people with special needs (e.g., nutrient-rich foods for the elderly) f) can be easily available and cost-effective
**Disadvantages of Nutraceuticals**
a) the choice of people in clinical trials is often so heterogeneous that variable results are obtained making difficult the comparison between the studies carried out b) clinical studies are often carried out with products that contain a mix of nutraceuticals, for which it is not always possible to determine the singular bioavailability or any synergistic, agonist or antagonist effects c) there are no diagnostic and prognostic scores that allow to verify the nutraceutical effect in a reproducible and comparable manner: the health effects of these products are generally verifiable only many years after administration

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
