# Peer review of "Functional Foods: An Approach to Modulate Molecular Mechanisms of Alzheimer’s Disease"

_cells, 2020, doi:10.3390/cells9112347_

Round 1

Reviewer 1 Report

Re: Manuscript ID: cells-956319.

I congratulate the authors on their well written review dealing with the intriguing relationship between diet and epigenetics, with special reference to Alzheimer’s disease. The different arguments are well balanced and explored. The authors have expertise in this field. However, some minor changes are suggested to improve the paper.

Minor points

1) line 4. Replace “Alzheimer” with “Alzheimer’s”.

2) Latin names of species must be written in italics (lines 585-589, 593, 716, 770-771, 876, 1027, 1145, 1147, 1210, 1222, 1232, 1249, 1251, 1252, 1260, 1345, 1350, 1369, 1373, 1377, 1383, 1391, 1392, 1395, 1398, 1403, 1418, 1419, 1423, 1424, 1427, 1484, 1487, 1498, 1768, 1770).

3) line 831. Replace “beta-carotene” with “β-carotene”.

4) line 1168. Replace “improvements” with “improvement”.

5) line 1303. Replace “endocrin” with “endocrine”.

6) line 1350. Replace “bifidobacteria” with “Bifidobacteria”.

7) line 1710. Replace “betain” with “betaine”.

8) line 1710. Replace “enzyme’s” with “enzyme”.

9) line 1719. Replace “Lucarelli’” with “Lucarelli’s”.

10) line 1739. Delete “of”.

11) line 1849. Replace “of” with “in”.

12) line 1857. Replace “3-b” with “3-β”.

13) line 1865. Replace “w-3” with “ω-3”.

14) line 1904. Replace “[394]” with “[395]”. Reference [394] appears in bold in the bibliography and is never cited in the text. Check the correct correspondence.

15) line 1924. Replace “Histone-” with “histone”.

16) line 1950. Replace “Alzheimer’s” with “Alzheimer’s disease”.

17) line 1951. Replace “it not” with “it is not”.

18) line 1982. Replace “Alzheimer’s” with “Alzheimer’s disease”.

Author Response

I congratulate the authors on their well written review dealing with the intriguing relationship between diet and epigenetics, with special reference to Alzheimer’s disease. The different arguments are well balanced and explored. The authors have expertise in this field. However, some minor changes are suggested to improve the paper.

We thank very much the reviewer. Our responses to the reviewer's comments are in BLU.

Minor points

MINOR POINTS - indicated by the Reviewer - have been corrected. However, please note that the line number indicated for each point does not correspond to the line number of the downloaded manuscript. For example, the replacement "improvements" with "improvement" - the reviewer indicates line 1168 - is on line 1176 of the manuscript we downloaded. In the NEW version, it is on line 1203 However we managed to track down and correct all minor points.

1) line 4. Replace “Alzheimer” with “Alzheimer’s”. Ok
2) Latin names of species must be written in italics (lines 585-589, 593, 716, 770-771, 876, 1027, 1145, 1147,
1210, 1222, 1232, 1249, 1251, 1252, 1260, 1345, 1350, 1369, 1373, 1377, 1383, 1391, 1392, 1395, 1398, 1403, 1418, 1419, 1423, 1424, 1427, 1484, 1487, 1498, 1768, 1770). Ok
3) line 831. Replace “beta-carotene” with “β-carotene”. Ok
4) line 1168. Replace “improvements” with “improvement”. Ok
5) line 1303. Replace “endocrin” with “endocrine”. Ok
6) line 1350. Replace “bifidobacteria” with “Bifidobacteria”. Ok
7) line 1710. Replace “betain” with “betaine”. Ok
8) line 1710. Replace “enzyme’s” with “enzyme”. Ok
9) line 1719. Replace “Lucarelli’” with “Lucarelli’s”. Ok
10) line 1739. Delete “of”. Ok
11) line 1849. Replace “of” with “in”. Ok
12) line 1857. Replace “3-b” with “3-β”. Ok
13) line 1865. Replace “w-3” with “ω-3”. Ok
14) line 1904. Replace “[394]” with “[395]”. Reference [394] appears in bold in the bibliography and is never cited in the text. Check the correct correspondence.

Referee is right. We apologize for the confusion.

Ref 394 has been eliminated

Old Ref 395 is the reference exact for Santa-Maria et al. It becomes 394.

New Ref 395 is Ravi et al., according to Referee 2's suggestion.

Ref 396 has been eliminated.

15) line 1924. Replace “Histone-” with “histone”. Ok
16) line 1950. Replace “Alzheimer’s” with “Alzheimer’s disease”. Ok
17) line 1951. Replace “it not” with “it is not”. Ok
18) line 1982. Replace “Alzheimer’s” with “Alzheimer’s disease”. Ok

Reviewer 2 Report

The manuscript by Atlante and colleagues proposes to review the possible use of nutraceuticals as anti-Alzheimer’s disease therapeutics.  This review is certainly timely and of importance in the field, well written and easy to read. However, several issues have to be solve before publication:

#1- The manuscript is too long and would benefit from a shortening. Some paragraphs are redundant (3.1/4.2; 3.2/4.3; 3.3/4.1) and should be merged. Moreover, the impact of nutraceuticals on epigenetic changes (as announced in the title) represents only one part of the review (section 6). Therefore, either the title should be simplified to “Functional foods: An approach to modulate molecular mechanisms of Alzheimer’s disease” or the manuscript should exclusively focus on that aspect.

#2- Surely more figures/tables would help summarizing/visualising all the data discussed in the text.

#3- The authors should be careful when mentioning previous experimental works but citing review articles instead. In other words, statements referring to original findings should be attributed accurately. For instance, page 19, line 900, the authors wrote: “In particular, Ahmed et al. [198] showed that……” while this reference is a review article. The same remark applies to page 20, line 908-909 (“Another study, having observed that RSV effectively reduces the cleavage-mediated activation of APP and promotes peptide clearance [201]”) with ref 201 being a review article. The original demonstration of a stimulation of Ab clearance by RSV was actually provided by Marambaud et al., J. Biol. Chem. 2005 280, 37737-37382. Again, page 20 line 926, “Based on the opposite findings of Marx et al [204], which reported that…..” refers to a review article and not to the seminal experimental work.

#4- The recent review in the field “Neuro-nutrients as anti-Alzheimer’s disease agents: A critical review” by Ravi et al. 2019 Crit. Rev. Food Sci Nutr. 18, 2999-3018 should be cited.

#5- Some scientific inaccuracies appear along the manuscript:

-  Page 10 line 398 and 429-434: “To date, Ab and Tau are competing for the role of the main responsible factor for the disease". This debate is no longer really relevant since the recent demonstration that both factors are tightly linked and indeed act synergistically (see Ittner et al. 2018 Neuron 99, 13-27 for review).

- Page 10, line 401: The toxic (not always) Ab peptide derives from the proteolytic cleavage of APP by b-secretase AND g-secretase

- The relatively recent but widely accepted fact that Ab oligomers are the molecular entities mainly responsible for neuronal death should be mentioned and discussed.

Round 2

Reviewer 2 Report

The authors have significantly improved the manuscript following the reviewer's recommendations.